# Genetics of *trans*-regulatory variation in gene expression

Frank Wolfgang Albert[1†*], Joshua S Bloom[2,3,4†*], Jake Siegel[2,3,4], Laura Day[2,3,4], Leonid Kruglyak[2,3,4*]

[1]Department of Genetics, Cell Biology and Development, University of Minnesota, Minneapolis, United States; [2]Department of Human Genetics, University of California, Los Angeles, Los Angeles, United States; [3]Department of Biological Chemistry, University of California, Los Angeles, Los Angeles, United States; [4]Howard Hughes Medical Institute, Los Angeles, United States

**Abstract** Heritable variation in gene expression forms a crucial bridge between genomic variation and the biology of many traits. However, most expression quantitative trait loci (eQTLs) remain unidentified. We mapped eQTLs by transcriptome sequencing in 1012 yeast segregants. The resulting eQTLs accounted for over 70% of the heritability of mRNA levels, allowing comprehensive dissection of regulatory variation. Most genes had multiple eQTLs. Most expression variation arose from *trans*-acting eQTLs distant from their target genes. Nearly all *trans*-eQTLs clustered at 102 hotspot locations, some of which influenced the expression of thousands of genes. Fine-mapped hotspot regions were enriched for transcription factor genes. While most genes had a local eQTL, most of these had no detectable effects on the expression of other genes in *trans*. Hundreds of non-additive genetic interactions accounted for small fractions of expression variation. These results reveal the complexity of genetic influences on transcriptome variation in unprecedented depth and detail.
DOI: https://doi.org/10.7554/eLife.35471.001

**\*For correspondence:**
falbert@umn.edu (FWA);
jbloom@mednet.ucla.edu (JSB);
LKruglyak@mednet.ucla.edu (LK)

[†]These authors contributed equally to this work

## Introduction

Differences in gene expression among individuals arise in part from DNA sequence differences in regulatory elements and in regulatory genes. Regions of the genome that contain regulatory variants can be identified by tests of genetic linkage or association between mRNA levels and DNA polymorphisms in large collections of individuals. Regions for which such tests show statistical significance are known as eQTLs (*Albert and Kruglyak, 2015*). Regulatory variation is widespread in the species for which it has been studied; indeed, in humans, the expression of nearly every gene appears to be influenced by one or more eQTL (*Aguet et al., 2017*; *Battle et al., 2014*).

In humans, eQTLs are typically mapped by genome-wide association studies (GWAS) in unrelated individuals. To cover the genome, human GWAS must test a very large number of variants, resulting in a high multiple-testing burden and low statistical power. As a result, most human eQTL GWAS have been limited to searches for 'local' eQTLs that are located close to the genes they influence (*GTEx Consortium, 2015*; *Lappalainen et al., 2013*). The power to detect local eQTLs is higher because focused local tests reduce the multiple-testing burden, and because local eQTLs tend to have larger effect sizes. However, genome-wide estimates show that most regulatory variation does not arise from local eQTLs. Instead, it arises from 'distant' eQTLs, which are located far from the genes they influence, typically on different chromosomes, and which exert their effects through *trans*-acting factors (*Grundberg et al., 2012*; *Wright et al., 2014*). Although *trans*-acting human eQTLs have been discovered (*Aguet et al., 2017*; *Battle et al., 2014*; *Brynedal et al., 2017*; *Fehrmann et al., 2011*; *Grundberg et al., 2012*; *Heinig et al., 2010*; *Lee et al., 2014*; *Small et al.,*

**eLife digest** Every individual's genome is unique, with variations in the DNA sequence at many thousands of points. Each difference is a change in one or more 'letters' of the DNA code. Some of these DNA letter variations have consequences for the way the individual looks or behaves. They can influence these traits either by changing the sequence of a protein encoded by a gene; or by changing when, where or how much a gene is active.

Studying how individual differences in the DNA influence gene activity requires a very large amount of data on many individuals within a species. Only recently have such large datasets become available. These have made it possible to study these regulatory differences in unprecedented detail.

Albert, Bloom et al. set out to map as many regulatory genetic variants as possible in budding yeast – a popular model organism used in many branches of science. The approach involved measuring how active every gene in the genome was, and which genetic variants influenced whether each gene's activity was turned up or down, in more than 1,000 different strains of yeast. Thousands of regions of the DNA turned out to influence regulation of genes. The analysis revealed that almost every gene is influenced by a complex set of regulatory regions all over the genome. Some hotspot regions were found to influence thousands of genes at once.

The findings provide the most complete set of data for studying the effects of variation in DNA sequence on genetic regulation in any species, and can act as a model for researchers to carry out similar experiments in other species. Ultimately, these results could help understand exactly how differences in genome sequence help to make individuals unique.

DOI: https://doi.org/10.7554/eLife.35471.002

*2011*; *Wright et al., 2014*; *Yao et al., 2017*), the vast majority remains unknown. As a consequence, we know relatively little about this crucial source of regulatory genetic variation.

In model organisms, eQTLs can be identified by linkage analysis in panels of offspring obtained from crosses of genetically different individuals (*Brem et al., 2002*). Whereas GWAS studies are powered to test only genetic variants found at high frequency in the population (e.g. [*Kita et al., 2017*]), linkage studies can assay both common and rare variants that differ between the parental strains. In addition, longer blocks of linkage reduce the number of statistical tests required to cover the genome. As a result, many local and distant eQTLs have been discovered in such studies. However, even in linkage studies, sample size limitations have to date resulted in insufficient statistical power to detect most eQTLs. This limitation has manifested itself as 'missing heritability': detected eQTLs tend to account for only a fraction of the measured heritable component of gene expression variation. Here, we addressed this limitation by carrying out an eQTL study in a large panel of segregants from a cross between two yeast strains. The high power of our study allowed us to identify eQTLs that account for the great majority of heritable expression variation in this cross, and to characterize the distant component of regulatory variation in unprecedented depth and detail.

## Results

### Deep eQTL mapping explains most gene expression heritability

We developed an experimental pipeline for high-throughput generation of RNA-seq data in yeast and obtained high-quality expression measurements (*Source data 1* and *Source data 2*) for 5720 genes in 1012 segregants from a cross between a laboratory and a wine strain (hereafter, BY and RM, respectively). We obtained high-confidence genotypes at 11,530 variant sites from low-coverage whole-genome sequences of the segregants (*Bloom et al., 2013*) (*Source data 3*). We used the genotype and RNA-seq data for eQTL mapping and identified 36,498 eQTLs for 5643 genes at a false discovery rate (FDR) of 5% (*Source data 4*). Only 77 genes had no detected eQTL. Among the genes with at least one detected eQTL, the median number was 6, with a maximum of 21 (*Figure 1A*; Supplementary Discussion 1 describes the five genes with 21 eQTLs). Previous eQTL mapping in 112 segregants from this cross detected an average of less than one eQTL per gene as a consequence of much lower statistical power (*Brem et al., 2002*; *Smith and Kruglyak, 2008*). That

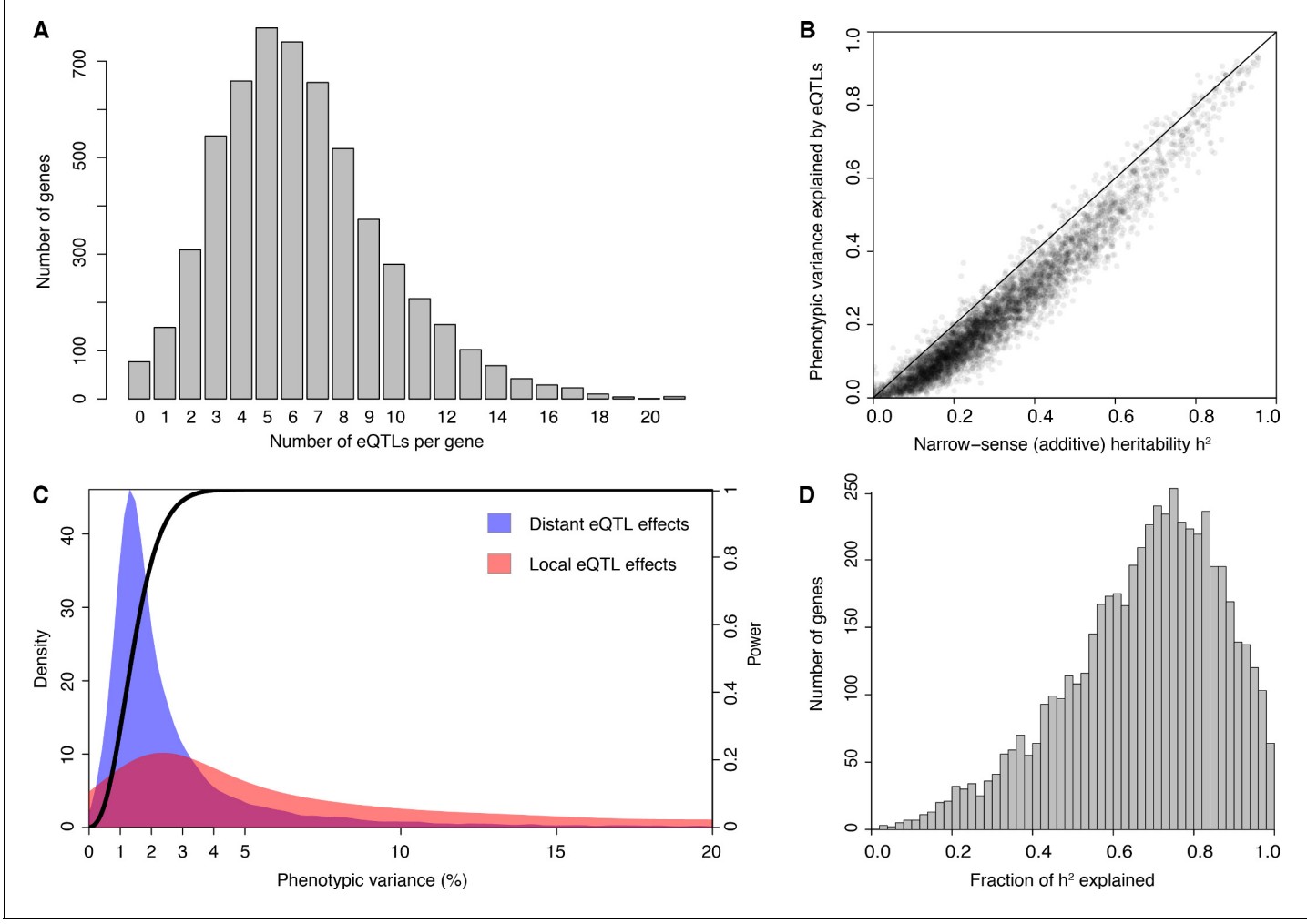

**Figure 1.** eQTL detection and transcriptome heritability. (A) Histogram showing the number of eQTLs per gene. (B) Most additive heritability for transcript abundance variation is explained by detected eQTLs. The total variance explained by detected eQTLs for each transcript (y-axis) is plotted against the additive heritability ($h^2$). The diagonal line represents a scenario under which the variance explained by eQTLs exactly matches the heritability. (C) Power to detect eQTLs as a function of effect size, and distributions of observed local and distant eQTL effects. The black curve corresponds to the statistical power (right y-axis) for eQTL detection at a genome-wide significance threshold. Colored areas show the density of individual significant eQTLs (left y-axis) that explain a given fraction of phenotypic variance (x-axis) for distant (blue) and local (red) eQTLs. Note that the x-axis is truncated at 20% variance explained to aid visualization of smaller effects, and omits a long tail of few eQTLs with large effects. (D) Histogram showing the fraction of $h^2$ explained by the sum of the eQTLs for each gene.

DOI: https://doi.org/10.7554/eLife.35471.003

The following figure supplements are available for figure 1:

**Figure supplement 1.** Mean additive heritability across all transcripts as a function of downsampling the total number of reads per sample.

DOI: https://doi.org/10.7554/eLife.35471.004

**Figure supplement 2.** Heritability ($h^2$) compared to other measures.

DOI: https://doi.org/10.7554/eLife.35471.005

data set was used to obtain indirect estimates of the distribution of the number of eQTLs per gene (**Brem and Kruglyak, 2005**), and these agree closely with the distribution of directly detected eQTLs observed in the current study. For example, Brem and Kruglyak (**Brem and Kruglyak, 2005**) estimated that at most 3% of genes would be influenced by a single eQTL (we observed 2.6% of such genes), and half of genes would require >5 eQTLs (we observed a median of 6 eQTLs per gene). While Brem and Kruglyak estimated that one third of genes would require more than 8 eQTLs, we observed only 23% such genes. Additional eQTLs of very small effect missed by our study likely account for this discrepancy. The observed distribution of the number of loci also closely matched

the distribution we reported for loci influencing 160 protein levels studied with the highly powered X-pQTL approach (*Albert et al., 2014b*). Our results provide direct demonstration that variation in expression levels of nearly all genes has a complex genetic basis.

We used our data to estimate the additive heritability of the expression level of each gene (i.e. the fraction of expression variance attributable to genetic factors; *Figure 1—figure supplement 1*; *Source data 5*). We observed a median heritability of 26%, with a maximum of 95% (*Figure 1B*). Our estimates are similar to those from population-based studies of gene expression in humans (*Grundberg et al., 2012*; *Lloyd-Jones et al., 2017*; *Wheeler et al., 2016*; *Wright et al., 2014*). The estimates are lower than heritabilities typically seen for organismal traits in this yeast cross (*Bloom et al., 2013*, *2015*), suggesting a greater contribution of environmental and stochastic factors to gene expression variation. Across genes, heritability was positively correlated with mean expression and with expression variance, and negatively correlated with the number of protein-protein and synthetic genetic interaction partners, as well as with gene essentiality (p≤0.005) (*Figure 1—figure supplement 2*; Supplementary Discussion 2 and 3; *Supplementary file 1*; *Source data 6*).

In contrast to previous eQTL studies, the detected eQTLs explained most of the estimated additive gene expression heritability (a median across genes of 71.5%) (*Figure 1B and D* 10-fold cross-validation). Low missing heritability in our data is explained by the high power of our experiment. We had greater than 90% power to detect eQTLs that explain at least 2.5% of expression variance (*Figure 1C*). The distribution of effect sizes of detected eQTLs is strongly weighted toward small effects (median 1.9% of variance explained; *Figure 1C*), suggesting that the remaining missing heritability is explained by undetected eQTLs with even smaller effects. These results are similar to those observed for organismal traits in this cross (*Bloom et al., 2013*, *2015*). Thus, we have discovered most eQTLs with substantial effects that segregate in this cross, and these jointly account for the great majority of the observed genetic variation in the transcriptome.

## Genetic expression variation arises primarily from *trans*-acting hotspots

We found that 2884 genes (50% of 5720 expressed genes) had a local eQTL (defined as an eQTL whose confidence interval includes the gene it influences) at genome-wide significance (*Figure 2A*). This number rose to 4241 genes (74% of expressed genes) when we performed eQTL analysis with

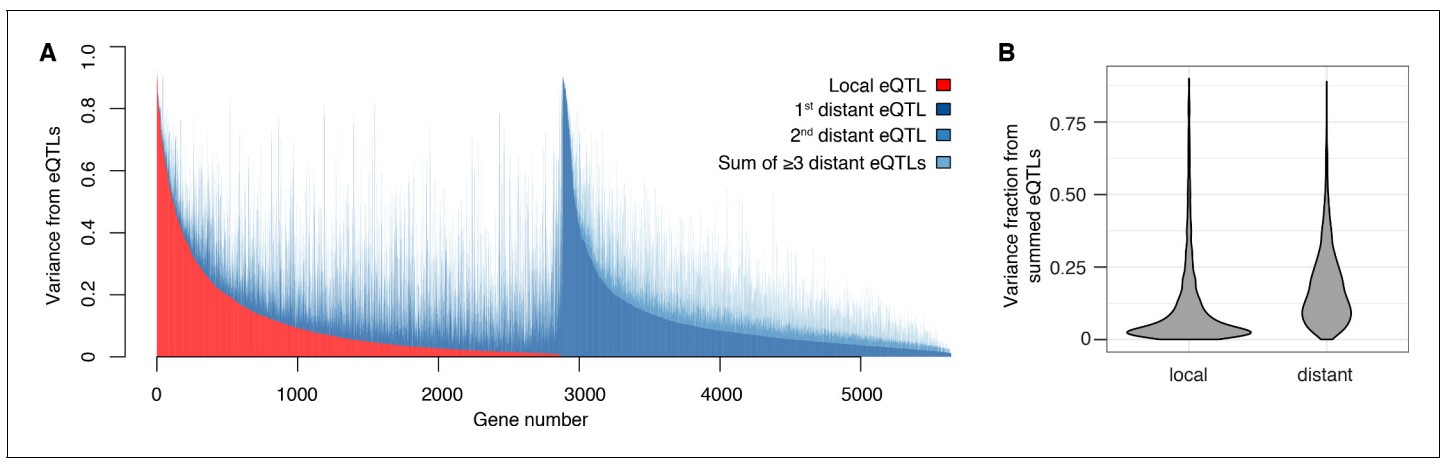

**Figure 2.** Contribution of local and distant eQTLs to expression variance. (**A**) A stacked barplot showing for each gene with at least one eQTL the amount of phenotypic variance from local and distant eQTLs. Genes are sorted first by the amount of variance from the local eQTL, followed by the amount of variance from the strongest distant eQTL. (**B**) Violin plots of the distributions of fractions of phenotypic variance explained by summed local and distant eQTLs, respectively. This panel was generated using genes with at least one local and one distant eQTL.

DOI: https://doi.org/10.7554/eLife.35471.006

The following figure supplements are available for figure 2:

**Figure supplement 1.** Allele-specific expression (ASE) compared to local eQTL effects.
DOI: https://doi.org/10.7554/eLife.35471.007

**Figure supplement 2.** Power to detect allele-specific expression.
DOI: https://doi.org/10.7554/eLife.35471.008

only one nearby marker per gene in order to reduce the multiple testing burden (FDR < 5%). Thus, the single pair of yeast isolates used here harbors sufficient local regulatory variation to alter the expression of more than half the genes in the genome. Comparisons with allele-specific expression data (*Albert et al., 2014a*) support previous results (*Doss et al., 2005*; *Ronald et al., 2005*) that most but not all local eQTLs act in *cis* (*Figure 2—figure supplement 1*, *Figure 2—figure supplement 2*, Supplementary Discussion 4; *Supplementary files 2*, *3* and *4*; *Source data 7*).

The vast majority of the genome-wide significant eQTLs did not overlap the genes they influenced (92%; 33,529 of 36,498); indeed, 86% were located on a different chromosome. Nearly every expressed gene (98%; 5606) had at least one such distant, *trans*-acting eQTL (*Figure 2A*). The individual effect sizes of the *trans* eQTLs were smaller than those of local eQTLs (median variance explained 2.8-fold less, T-test p<2.2e-16; *Figure 1C*). However, for the 2846 genes that had both a local eQTL and at least one distant eQTL, the aggregate effect of the distant eQTLs per gene was larger than that of the local eQTL (median 2.6-fold more variance explained; paired T-test p<2.2e-16; *Figure 2B*). Distant eQTLs accounted for the majority of eQTL variance for 85% of genes. Our results directly demonstrate the importance of *trans* acting variation.

The *trans* eQTLs were not uniformly distributed across the genome (*Figure 3A*). Instead, they clustered at 102 hotspot loci, each of which affected the expression of many genes (*Brem et al., 2002*; *Ghazalpour et al., 2008*; *Orozco et al., 2012*) (*Figure 3B*). These hotspots contained over 90% of all *trans* eQTLs. The eQTLs that mapped outside of the hotspots also clustered more than expected by chance (randomization p<0.001), suggesting the existence of additional hotspots that affect the expression of too few genes to pass the stringent criteria used to define the set of 102. Isolated *trans*-acting loci that affect the expression of one or a few genes appear to be uncommon.

The 102 hotspots affected a median of 425 genes, ranging from 26 (a newly discovered hotspot at position 166,390 bp on chromosome III) to 4594 at the previously reported *MKT1* hotspot (*Zhu et al., 2008*) (82% of 5629 genes with any signal at a hotspot; *Figure 3B*). Three additional hotspots each affected more than half of all genes. They include a previously described hotspot at the *HAP1* gene (*Brem et al., 2002*) (3640 genes affected), as well as two newly detected hotspots on chromosome XIV. A hotspot at 372,376 bp affected 4172 genes and is likely caused by a variant that recently arose in the *KRE33* gene in the RM parent used in our cross (*Jerison et al., 2017*). A hotspot at position 267,161 bp affected 3169 genes and spans the genes *GCR2*, YNL198C, *WHI3* and *SLZ1*. These results indicate that hotspots can have extraordinarily wide-reaching effects on the transcriptome, with some influencing the expression of the majority of all genes.

Widespread effects caused by single loci likely arise from a cascade of effects in which strong primary effects spread through the cellular regulatory network. For example, the BY allele of the transcriptional activator *HAP1* carries a transposon insertion that reduces *HAP1* function. As expected, the BY allele strongly reduced the expression of known transcriptional targets of *HAP1*: 26 out of the 69 *HAP1* targets present in our data were among the 50 genes with the largest reduction in expression in segregants carrying the BY allele of *HAP1* (p<2.2e-16, odds ratio = 138). In total, only 75 direct transcriptional *HAP1* targets are known. Unless previous work missed thousands of *HAP1* targets, the vast majority of the 3640 *trans* eQTLs at *HAP1* must reflect indirect, secondary consequences of the direct transcriptional effects. *HAP1* is an activator of genes involved in cellular respiration. Thus, the many secondary effects of the BY *HAP1* allele on gene expression may be mediated by cellular responses to altered metabolism arising from reduced respiration.

Our Supplementary Files and Datasets provide detailed information about each hotspot. *Source data 8* contains a table that gives an overview of the hotspots, including their location, genes affected (details in *Source data 9*), and analyses of function (details in *Source data 10*) and transcriptional regulation (details in *Source data 11*) of the target genes of each hotspot. *Supplementary file 5* visually represents each hotspot region, and *Supplementary file 6* displays gene networks formed by the strongest target genes of each hotspot.

## Causal genes underlying hotspots

Functional analysis of eQTL hotspots requires identification of the underlying causal genes, which has been challenging to do systematically. We developed a multivariate fine-mapping algorithm that narrows hotspot positions by leveraging information across the genes that map to each hotspot (Materials and methods). Briefly, we used all genes with an eQTL on a given chromosome and regressed out genetic factors on all other chromosomes and additional non-genetic factors. We

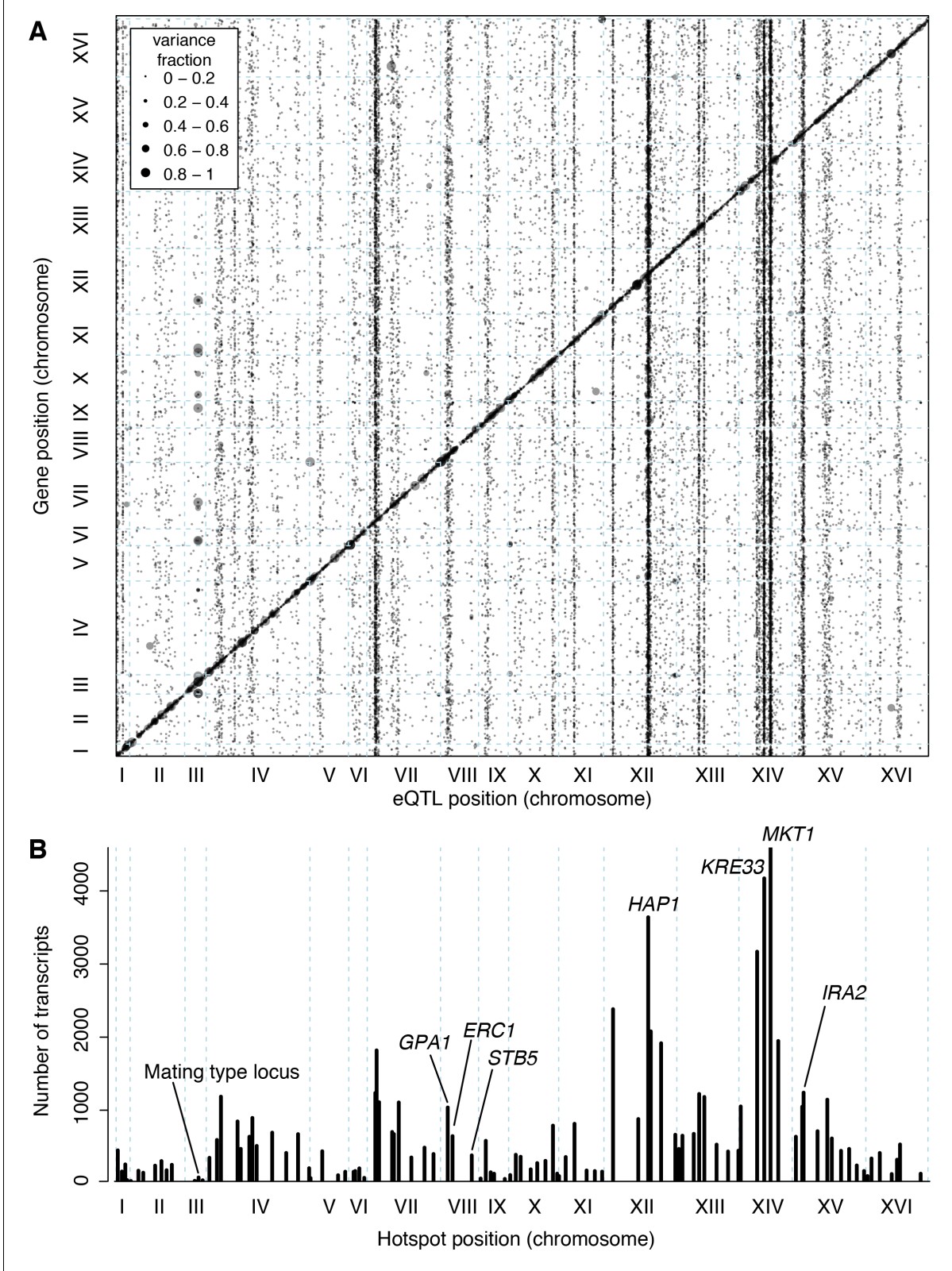

**Figure 3.** Locations of eQTLs in the genome. (**A**) Map of local and distant eQTLs. The genomic locations of eQTL peaks (x-axis) are plotted against the genomic locations of the genes whose expression they influence (y-axis). The strong diagonal band corresponds to local eQTLs. The many vertical bands correspond to eQTL hotspots. Point size is scaled as a function of eQTL effect size, measured in fraction of phenotypic variance explained. (**B**)

*Figure 3 continued on next page*

*Figure 3 continued*

The number of gene expression traits linking to each of 102 identified eQTL hotspots (Methods) are shown as vertical bars. Text labels identify genes in hotspots referred to in the text.

DOI: https://doi.org/10.7554/eLife.35471.009

reduced the dimensionality of the residual expression levels by singular value decomposition to capture linear combinations of traits that account for most of the variance in residual expression. We scanned the given chromosome for genetic influences on the multivariate distribution of these linear combinations, while controlling FDR via permutations. Finally, we used bootstraps to compute confidence intervals for hotspot locations (*Figure 4 and 5*).

With this approach, we resolved the locations of 26 hotspots to regions containing three or fewer genes (a total of 58 genes; *Figure 4A*). Three hotspots contained exactly one gene (*GIS1*, *STB5*, and *MOT3*). We previously identified and experimentally confirmed the causal genes at several major hotspots (*MKT1* (*Zhu et al., 2008*), *HAP1* (*Brem et al., 2002*), *IRA2* (*Smith and Kruglyak, 2008*), *GPA1* (*Yvert et al., 2003*), and the mating-type locus [*Brem et al., 2002*]). These were all correctly localized by the algorithm, validating this fine-mapping strategy.

The 58 genes at 26 high-resolution hotspots are highly enriched for the causal genes underlying the hotspots, making it possible to systematically study the functions of hotspot regulators. These genes were less likely to be essential or to have a human homolog than other yeast genes but did not differ from other genes in their expression level or the number of physical or genetic interaction partners (*Supplementary file 7*). The hotspot genes had highly significant enrichments for gene ontology (GO) terms related to transcriptional regulation (e.g. GO:0006357 'regulation of transcription from RNA polymerase II promoter': 19 genes among the 58; 4 expected; p=4e-9; *Figure 4—figure supplement 1*; *Source data 12*), as well as weaker enrichments for terms related to response to nutrient levels (GO:0031669; 'cellular response to nutrient levels': 8 genes, 1 expected, p=6e-6)

These analyses indicate that causal hotspot genes are disproportionately involved in transcriptional regulation, a signal that was not picked up in an earlier study with fewer, less-well-resolved hotspots (*Yvert et al., 2003*). For example, we fine-mapped a new hotspot that affected 382 genes to a single gene, the transcription factor *STB5* (*Figure 4B*). *STB5* is a transcriptional activator of multidrug resistance genes (*Kasten and Stillman, 1997*). A previous analysis suggested reduced activity of the *STB5* BY allele compared to the RM allele (*Lee and Bussemaker, 2010*). Consistent with this observation, we found that the promoters of genes whose expression was lower in the presence of the *STB5* BY allele were strongly enriched for *STB5* binding sites (*Figure 4C*).

To further examine the role of sequence variation in transcription factor genes, we focused on transcription factors with variants predicted to be damaging to protein function such as premature stop codons or frameshifts. There are eight such genes in our data. Remarkably, six of these (*GAT1*, *HMS1*, *PUT3*, *RFX1*, *SRD1*, *TBS1*) were located in a hotspot, often very close to the estimated peak location (*Figure 4—figure supplement 2*). None of these expression hotspots have been reported previously, although variation at *GAT1* has been reported to influence traits relevant for wine production (*Salinas et al., 2012*), and a premature stop codon in the RM allele of *RFX1* has been linked to reduced activity of this transcriptional repressor (*Lee and Bussemaker, 2010*). The remaining two transcription factors with predicted damaging mutations did not overlap a hotspot. Of these, a predicted frameshift in *YRM1* is very close to the end of the coding region, while a predicted frameshift in *STB4* appears to be an annotation artifact: it resides in a region that is annotated as coding but that does not in fact appear to be transcribed (*Figure 4—figure supplement 3*, [*Albert et al., 2014a*]).

Hotspot genes can also influence mRNA levels more indirectly – for instance, by shaping the cellular response to external stimuli such as nutrient availability. For example, we fine-mapped a hotspot, which influenced 645 genes, to an interval on chromosome VIII containing six genes (*Figure 4—figure supplement 4A*). One of these is *ERC1*, which encodes a transmembrane transporter. BY but not RM carries a frameshift in this gene, which removes the last two out of 12 predicted transmembrane helices of the protein (*Fehrmann et al., 2013*). This variant is known to reduce cell-to-cell variability (or 'noise') in the expression of a *MET17* gene tagged with green fluorescent protein (*Fehrmann et al., 2013*). We found that the BY allele at this hotspot reduced the

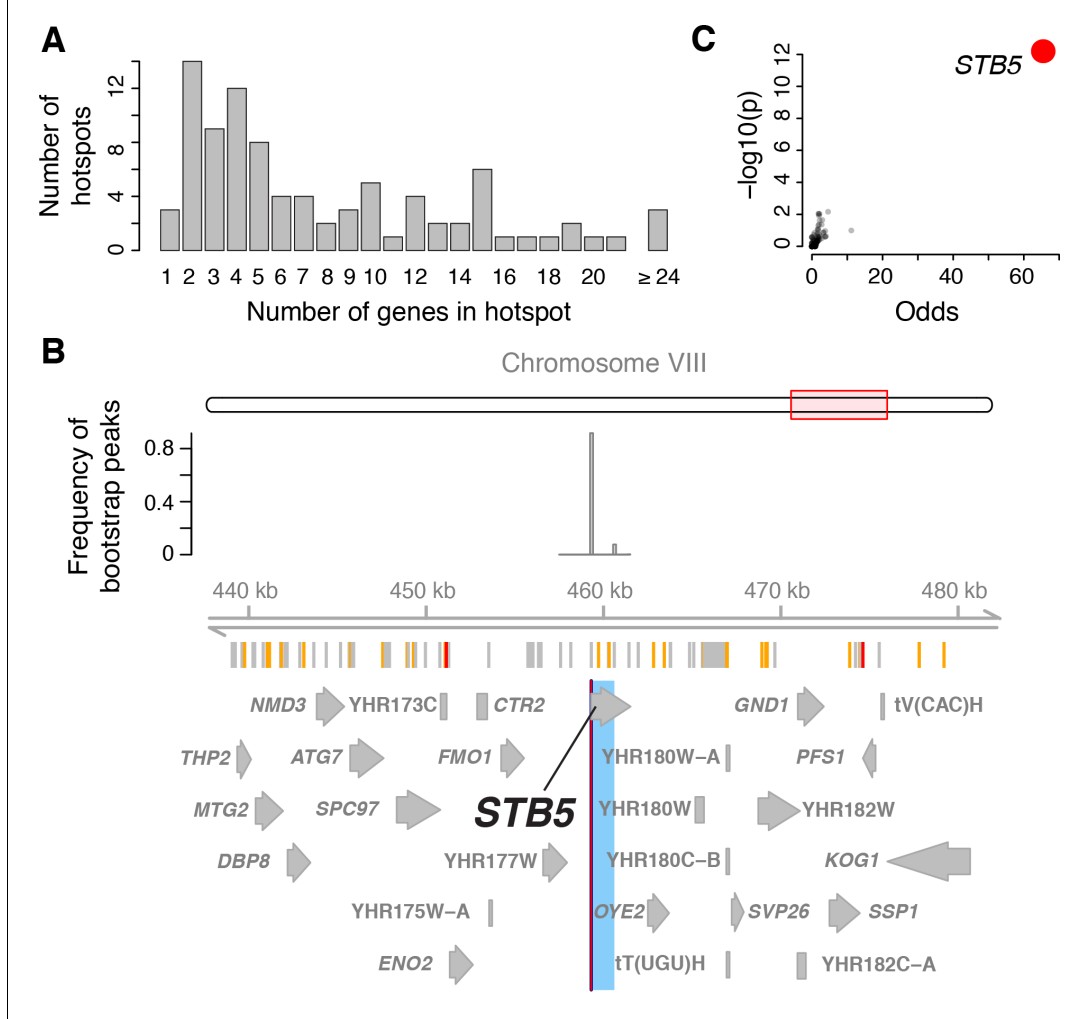

**Figure 4.** Genes located in hotspot regions. (**A**) Histogram showing the number of genes located in the hotspot regions. (**B**) A hotspot on chromosome VIII maps to the gene *STB5*. From top to bottom: the general region on the chromosome, the empirical frequency distribution of hotspot peak locations from 1000 bootstrap samples (Materials and methods), locations of BY/RM sequence variants (red: variants with 'high' impact such as premature stop codons (*McLaren et al., 2016*); orange: 'moderate' impact such as nonsynonymous variants; grey: 'low' impact such as synonymous or intergenic variants), and gene locations. The light blue area shows the 95% confidence interval of the hotspot location as determined from the bootstraps. The red line shows the position of the most frequent bootstrap marker. (**C**) Genes for which the BY allele at the *STB5* hotspot is linked to lower expression are enriched for *STB5* transcription factor (TF) binding sites in their promoter regions. The figure shows enrichment results for all annotated TFs (grey dots), with the strength of enrichment (odds ratio) on the x-axis vs. significance of the enrichment on the y-axis. The *STB5* result is highlighted in red.

DOI: https://doi.org/10.7554/eLife.35471.010

The following figure supplements are available for figure 4:

**Figure supplement 1.** Gene ontology (GO) enrichments of genes located in hotspots.

DOI: https://doi.org/10.7554/eLife.35471.011

**Figure supplement 2.** Hotspots at six transcription factor genes with damaging mutations.

DOI: https://doi.org/10.7554/eLife.35471.012

**Figure supplement 3.** mRNA and translation at *STB4*.

DOI: https://doi.org/10.7554/eLife.35471.013

**Figure supplement 4.** The *ERC1* hotspot.

DOI: https://doi.org/10.7554/eLife.35471.014

expression of genes that are highly enriched for the GO category 'methionine biosynthetic process' (GO:0009086, p=2e-22; *Source data 8* and *Source data 10*). Thus, in addition to reducing *MET17* expression noise, the *ERC1* frameshift variant is linked to reduced mean expression levels of multiple genes in the methionine biosynthesis pathway (the *MET* regulon; *Figure 4—figure supplement 4B*). While the precise compounds that are imported or exported by Erc1p are not known, the *ERC1* BY allele reduces cellular levels of *S*-Adenosylmethionine (SAM) (*Breunig et al., 2014*), a key component of methionine and cysteine amino acid metabolism (*Sadhu et al., 2014*). The *ERC1* BY allele may down-regulate the *MET* regulon via its effects on SAM, triggering further transcriptional changes in hundreds of genes.

## Relationship of local eQTLs and *trans* eQTLs

Most known causal variants underlying yeast eQTL hotspots are coding (*HAP1* (*Brem et al., 2002*), *MKT1* (*Zhu et al., 2008*), *GPA1*, *AMN1* (*Yvert et al., 2003*), *SSY1* (*Brown et al., 2008*); [*Fay, 2013*]); however, change in the expression of a *trans*-acting factor by a local eQTL is another plausible causal mechanism (*Sudarsanam and Cohen, 2014*; *Yao et al., 2017*). We found that a higher proportion of hotspots contained genes with a local eQTL than expected by chance (p=0.007; *Figure 5A*). The median effect size of the strongest local eQTL in these hotspots was larger than expected (p=0.003). These enrichments are consistent with some hotspots being caused by local eQTLs that alter the expression of a gene located at the hotspot position, which in turn leads to changes in the other transcript levels that map to the hotspot.

On the other hand, the majority of local eQTLs (60%) did not overlap any of the hotspots. Evidently, the expression changes caused by these local eQTLs did not in turn lead to detectable *trans* effects on many unlinked genes, within the limits of our statistical power.

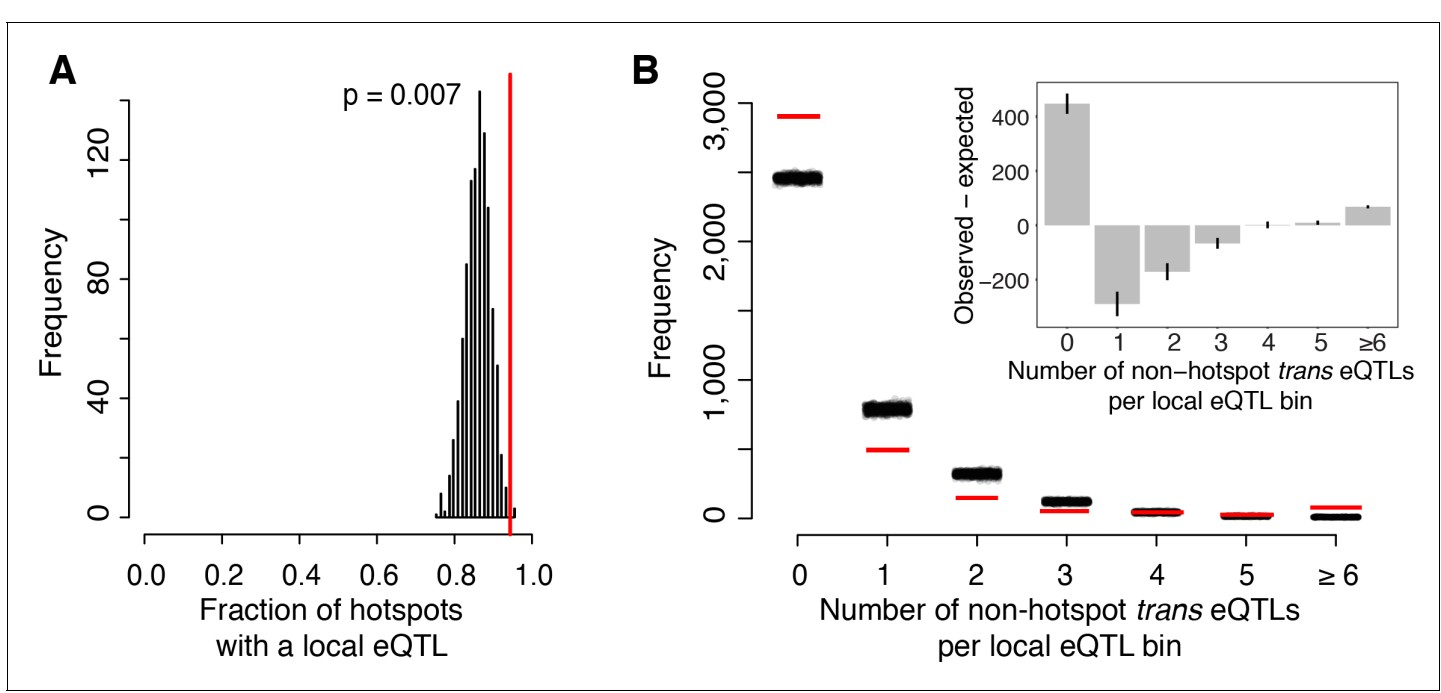

**Figure 5.** Relationship of local eQTLs and distant eQTL hotspots. (**A**) The fraction of hotspots that contain a genome-wide significant local eQTL. The black histogram shows the distribution observed in 1000 random, size-matched regions of the genome. Because of the high number of local eQTLs, most hotspots are expected to contain a local eQTL even by chance. The observed fraction (red line) still exceeds this random expectation. (**B**) Distribution of *trans* eQTLs at local eQTLs outside of hotspot regions. The genome was divided into non-overlapping bins centered on local eQTLs that did not overlap a hotspot. We counted the number of *trans*-eQTL peaks in each bin. The figure shows the frequency of bins with a given number of *trans*-eQTLs. The distribution observed in real data is shown by red lines, and distributions obtained in 1000 randomizations of *trans*-eQTL positions is show by clouds of black circles. The inset shows the observed less the expected frequency for each bin. Error bars indicate the 95% range from the randomizations.

DOI: https://doi.org/10.7554/eLife.35471.015

To quantify this observation further, we focused on 'non-hotspot' local eQTLs that did not overlap a hotspot and asked how they related to the 10% of non-hotspot *trans* eQTLs that did not overlap a hotspot. We created non-overlapping genomic bins, each centered on a non-hotspot local eQTL. We then counted how many non-hotspot *trans* eQTL peaks fell into these bins (*Figure 5B*). The resulting distribution roughly matched the distribution expected if non-hotspot *trans* peaks occurred at random locations. To the extent that the distribution differed from random, we found an excess of bins with six or more *trans* eQTLs (p<0.001). There was also an excess of bins with zero *trans* peaks (p<0.001). This class comprised the great majority of the distribution (*Figure 5B*). The genetic architecture that is most consistent with these observations is one in which some local eQTLs are accompanied by multiple *trans* eQTLs, while most local eQTLs have no detectable *trans* consequences on the expression of other genes.

Even when the causal gene in a hotspot has a local eQTL, it does not automatically follow that this is the causal mechanism. For example, *STB5* and *ERC1* each had a local eQTL. However, *STB5* did not show allele-specific expression, and while there was weak allele-specific expression for *ERC1*, it was in the opposite direction of the local *ERC1* eQTL (*Source data 7*). Therefore, these local eQTLs are unlikely to be caused by *cis*-acting variants.

*STB5* and *ERC1* carry protein-altering variants between BY and RM, including the known causal *ERC1* frameshift in BY. Altered protein activity due to these coding variants may be responsible for the many distant linkages to these hotspots and may also cause the observed local eQTLs in *trans*, as previously shown for *AMN1* (*Ronald et al., 2005*). The Stb5p transcription factor is predicted to target its own promoter (*MacIsaac et al., 2006*), such that its altered activity could influence its own expression. For the transmembrane transporter encoded by *ERC1*, the local eQTL might reflect a more indirect mechanism. For each of these hotspots, it seems plausible that a change in protein function, rather than change in gene expression, underlies the hotspot.

## Genetics of mRNA vs. protein levels

The degree to which mRNA-based eQTLs also affect the protein levels of their target genes is a fundamental open question (*Battle et al., 2015*; *Chick et al., 2016*; *Foss et al., 2007*; *Ghazalpour et al., 2011*; *Picotti et al., 2013*) that has been difficult to resolve as a consequence of low statistical power in eQTL and protein QTL (pQTL) studies. Low power is expected to lead to poor overlap between eQTLs and pQTLs solely as a result of high false-negative rates. We compared our eQTLs to pQTLs that we had identified earlier for 160 proteins using a powerful bulk segregant approach (*Albert et al., 2014b*) (*Source data 13*). Here, we present results comparing the distant QTLs in both datasets because – by design of our earlier study – the set of distant pQTLs is much larger than the set of local pQTLs. Results for local QTLs are broadly consistent with those for distant QTLs (Supplementary Discussion 5).

Distant pQTLs clustered at hotspots, which broadly mirrored the mRNA hotspots identified here (*Figure 6A*). However, differences in hotspot architecture exist. For example, a hotspot on chromosome II showed strong pQTL effects (*Albert et al., 2014b*) but only weak effects on mRNA levels for the same genes, none of which rose to genome-wide significance.

In order to avoid downward bias in the overlap between eQTLs and pQTLs caused by false negatives, we focused on strong QTLs in each dataset and asked if they overlapped a significant QTL in the other dataset (Materials and methods; see Supplementary Note 5 for results based on all distant QTLs). Of the 236 strongest eQTLs, 47% (111) overlapped a pQTL for the same gene. Of the 218 strongest pQTLs, 50% (108) overlapped an eQTL for the same gene. As a more sensitive alternative to QTL overlap analyses, we computed estimates of agreement based on the $\pi_1$ statistic (*Storey and Tibshirani, 2003*). Of the strong eQTLs, 92% were estimated to match a pQTL. Of strong pQTLs, 63% were estimated to match an eQTL. Thus, while nearly all eQTLs with strong effects on mRNA levels also affect protein levels for the same gene, a larger fraction of strong pQTLs appear to be specific to protein levels.

Strong eQTLs without a pQTL clustered primarily at the *HAP1* and *MKT1* hotspots (*Supplementary file 8*; *Figure 6B*). These two hotspots also showed the clearest examples of overlapping eQTLs and pQTLs with opposite direction of effect on the same genes (*Supplementary file 9*; *Figure 6C*). Thus, while these hotspots influence both mRNA and protein levels of many genes, their effects on mRNA vs. protein levels of a given gene can be quite different. Strong pQTLs without an eQTL were more widely distributed across the genome (*Supplementary file 10*; *Figure 6D*).

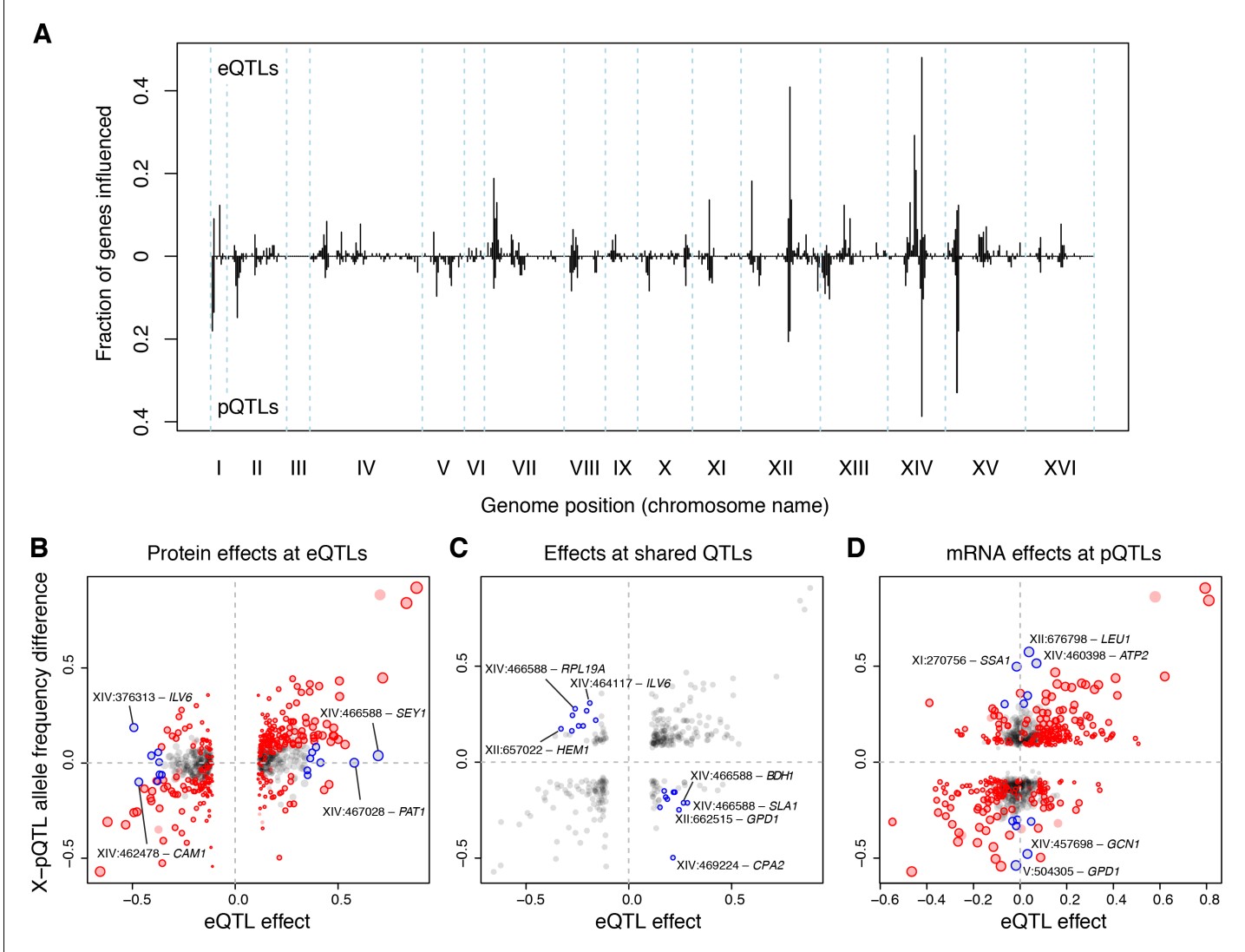

**Figure 6.** eQTLs and pQTLs. (**A**) Distant eQTL and pQTL hotspots. The figure shows the fraction of 154 genes (*Albert et al., 2014b*) that have an eQTL or pQTL in a given bin along the genome. eQTLs from the current dataset are shown in the upper half of the figure, and pQTLs from (*Albert et al., 2014b*) are shown in the bottom half with an inverted scale. Chromosome III is omitted from the figure because no pQTLs can be detected on this chromosome due to the experimental design of (*Albert et al., 2014b*). (**B–D**) Comparison of individual distant eQTLs and pQTLs. Each panel shows the effect size of linkage of mRNA levels for a given gene to a given genomic position (x-axis; correlation coefficient between mRNA level and marker genotype) compared to the effect size of linkage of protein levels for the same gene to the same genomic position (y-axis; difference in frequency of the BY allele between segregant pools with high and low expression of the protein [*Albert et al., 2014b*]). Positive values indicate higher expression in RM compared to BY. Only distant QTLs located on different chromosomes than their target gene are shown. (**B**) All distant eQTLs, irrespective of significance in pQTL data. Dot size scales as a function of eQTL effect size. Red circles: eQTLs that overlap a significant pQTL. Blue circles: strong eQTLs that do not overlap a pQTL (*Supplementary file 8*); extreme cases are indicated by QTL location and the name of the affected gene. (**C**) Overlapping significant eQTLs and significant pQTLs. Blue circles: overlapping QTLs with different direction of effect (*Supplementary file 9*); extreme cases are indicated. (**D**) All distant pQTLs, irrespective of significance in eQTL data. Dot size scales as a function of pQTL effect size. Red circles: pQTLs that overlap a significant eQTL. Blue circles: strong pQTLs that do not overlap an eQTL (*Supplementary file 10*); extreme cases are indicated.
DOI: https://doi.org/10.7554/eLife.35471.016

## Detection of non-additive eQTL interactions from a genome-wide search

The contribution of non-additive or 'epistatic' genetic interactions to trait variation is a topic of ongoing debate (*Hill et al., 2008*; *Mackay, 2014*; *Mäki-Tanila and Hill, 2014*). In particular, demonstration of non-additive effects on human gene expression has been challenging (*Becker et al.,*

*2012*; *Brown et al., 2014*; *Buil et al., 2015*; *Fish et al., 2016*; *Hemani et al., 2014*; *Wood et al., 2014*). Although clear examples of epistasis have been revealed for yeast gene expression (*Brem and Kruglyak, 2005*; *Storey et al., 2005*), the limited power of earlier studies had necessitated targeted search strategies rather than a full genome-by-genome scan.

We reasoned that the high power of our current dataset should permit a more unbiased view of the contribution of epistasis to mRNA expression variation. We carried out a genome-by-genome scan for non-additive interaction effects on the expression levels of all genes and detected 387 eQTL-eQTL interactions influencing 306 genes (FDR = 10%; *Source data 14*). To our knowledge, this is the first unequivocal identification of eQTL interactions from an unbiased genome-by-genome scan. Targeted scans with a reduced multiple testing burden identified larger numbers of interacting pairs of loci: a total of 784 from a scan for interactions between genome-wide significant additive eQTLs and the genome, and a total of 1464 interactions between significant additive eQTLs.

We examined the 387 eQTL-eQTL interactions detected in the genome-wide scan in more detail. The locations of interacting eQTLs clustered at certain positions in the genome, generally overlapping the hotspots described above (*Figure 7A*). In particular, many epistatic interactions involved the *HAP1* hotspot (79 interactions), the *KRE33* hotspot (66 interactions), as well as hotspots containing *MKT1*, *GAP1*, the mating type locus, and *IRA2*. Many interactions connected these hotspots with each other (*Figure 7B*). For example, 30 genes shared an eQTL interaction between *HAP1* and *KRE33*, 14 genes shared an eQTL interaction between *HAP1* and *MKT1*, 13 genes shared an eQTL interaction between *KRE33* with *IRA2*, and 14 genes shared eQTL interactions between *GPA1* and the mating-type locus (*Brem et al., 2005*).

The fact that interacting eQTLs colocalize with additive hotspots suggests that epistatic interactions often involve eQTLs that also have additive effects. Indeed, of the 774 markers in the 387 epistatic pairs, 558 (72%) were within 10 kb of a genome-wide significant additive eQTL influencing the same gene. An estimate based on the $\pi_1$ statistic (*Storey and Tibshirani, 2003*) showed that at least 84% of epistatic markers have additive effects. An example is *SAG1*, which encodes the Alpha-agglutinin of cells with the alpha mating type. In our cross, the alpha mating type was carried by the RM parent. Consequently, expression of *SAG1* was higher in segregants that carried the RM allele at the mating type locus (*Figure 7C*). In addition to this additive effect, the mating type locus was also involved in an interaction with the *GPA1* locus, replicating the finding from a targeted search that a BY-specific S469I variant in *GPA1* modulates *SAG1* expression in alpha but not **a** cells (*Brem et al., 2005*).

Of the 387 interactions, 158 (41%) involved '*cis* by *trans*' interactions between distant and local loci (*Figure 7—figure supplement 1A*). For example, the *HAP1* hotspot interacted with local eQTLs for 15 genes (*Figure 7—figure supplement 1A*). Among these, the RM alleles of *HAP1* and *SCM4* both increased *SCM4* expression, and segregants with the RM genotype at both loci had higher expression levels than expected from additive effects (*Figure 7—figure supplement 1B*). The *HAP1* BY allele is less active due to a transposon insertion (*Brem et al., 2002*). The local *SCM4* eQTL may arise from variants that disrupt a *HAP1* binding site (*Ter Linde and Steensma, 2002*) in BY, and this allelic difference may result in a stronger expression difference in the presence of the more active RM Hap1p. In agreement with this model, an epistatic interaction between inferred Hap1p activity and *SCM4* expression has been reported (*Parts et al., 2011*).

Interestingly, *SCM4* is the only annotated direct *HAP1* target among the local eQTLs that interact with *HAP1* (*Harbison et al., 2004*). Further, the effect sizes of local eQTLs that interacted with *HAP1* were not statistically different in segregants that carried the more active RM vs. the less active BY allele at *HAP1* (T-test, p=0.9). In all the cases we examined (*KRE33*, *MKT1*, *GPA1*, and *IRA2*), the local eQTLs that interacted with these hotspots did not show consistently larger or smaller effect sizes depending on the hotspot allele (all p≥0.3). This argues against a scenario in which most *cis* by *trans* interactions are due to variants in transcription factor binding sites whose effect increases with a more active *trans* regulator. Instead, most *cis* by *trans* interaction effects may be mediated through more indirect mechanisms.

We quantified the fraction of variation in gene expression that is contributed by epistatic interactions. Pairwise interactions typically explained about 1/10th as much expression variance as did additive loci (*Figure 7—figure supplement 2*), with a median of 2.2% of expression variance per pair. Thus, genetic interactions contributed only a small minority of trait variance for gene expression

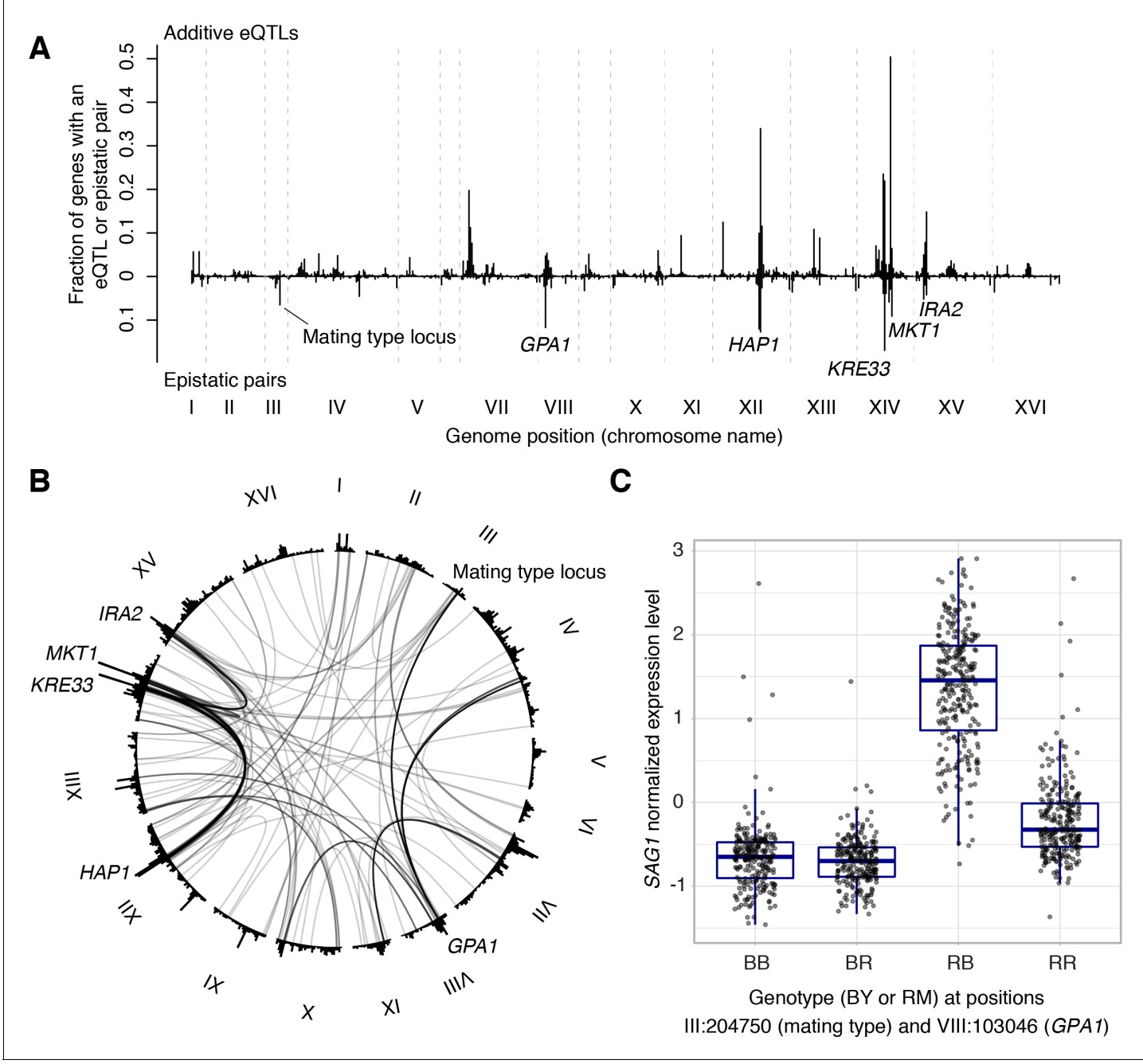

**Figure 7.** Non-additive interactions between eQTLs. (A) Locations of markers of epistatic pairs (pointing downward) compared to those of additive eQTLs (pointing upward). Epistatic hotspots discussed in the text are highlighted. (B) Interactions between two *trans* loci. The plot shows the genome broken up into chromosomes (indicated as roman numerals), with arches connecting two interacting loci. Arches are shaded such that multiple overlapping interactions appear darker. Epistatic hotspots are indicated as in panel A. The outer histogram shows the density of additive eQTLs. (C) Expression levels of *SAG1* as a function of genotypes at the mating type locus and *GPA1*.

DOI: https://doi.org/10.7554/eLife.35471.017

The following figure supplements are available for figure 7:

**Figure supplement 1.** *Cis* by *trans* interactions.

DOI: https://doi.org/10.7554/eLife.35471.018

**Figure supplement 2.** Distribution of the fraction of phenotypic variance (y-axis) explained by genetic variation as captured by genome-wide relatedness in an additive (left), and interactive (right) model.

DOI: https://doi.org/10.7554/eLife.35471.019

**Figure supplement 3.** Epistatic interactions without additive effects.

*Figure 7 continued*

DOI: https://doi.org/10.7554/eLife.35471.020

levels, which is consistent with what we previously reported for organism-level traits (*Bloom et al., 2015*).

We asked if any of the epistatic pairs identified from our unbiased search corresponded to 'purely' epistatic interactions without any additive effects. Of the 387 pairs, 111 (29%) were not found in the search between additive loci. Given the small effects of most interactions, this group is likely enriched for false positives from the full search as well as false negatives from the search between additive loci. We searched for specific epistatic pairs in which neither locus had an additive effect (p>0.05), overlapped an additive hotspot (to avoid any potential undetected additive hotspot effects), and where the affected gene was on a different chromosome from both interacting markers (to avoid any potential undetected additive local eQTLs). Only three such pairs existed in our data (*Figure 7—figure supplement 3*). In each case, the two genotype classes with non-parental geno-type combinations (BY-RM and RM-BY) had similar expression levels that differed from those in the two parental genotype combinations (BY-BY and RM-RM), such that the marginal genotype effects canceled out. Even in these three cases, the variance resulting from the interaction was only 2.6– 3.6%. We conclude that any contribution of 'purely' epistatic pairs to transcriptome variation must be small.

## Discussion

The high power of our study allowed us to identify genome-wide significant eQTLs that jointly explain over 70% of gene expression heritability. Thus, our eQTL map shows a high degree of com-pleteness, capturing most genetic sources of transcriptome variation in this cross. In particular, our results allowed us to examine the contribution of *trans*-acting regulatory variation in much greater detail than previously possible in any species.

We showed that *trans*-acting eQTLs are the predominant source of expression variation in this cross. Specifically, *trans*-eQTLs contribute 2.6-fold more to gene expression variance than local eQTLs, remarkably similar to genome-wide estimates in humans of 1.8-fold (*Grundberg et al., 2012*) and 3.4-fold (*Wright et al., 2014*). While based on different experimental designs and study populations, these results from yeast and humans clearly demonstrate the importance of *trans*-acting variation in eukaryotic species.

The vast majority of *trans*-eQTLs are concentrated at a limited number of hotspot regions that harbor variants with widespread effects on the expression of other genes. The strongest of these hotspots affected the expression of most genes in the genome. The minority of *trans* eQTLs that fell outside the statistically defined hotspots also clustered more than expected by chance. We saw little evidence for isolated *trans*-acting loci that affect the expression of one or a few genes. The high density of sequence differences in this cross might have been expected to produce a more diffuse *trans*-regulatory architecture, with many loci each influencing one or a few genes. Instead, *trans*-act-ing regulatory variation arises primarily from several dozen specific loci.

The large number and fine resolution of our hotspots allowed us to conduct unbiased analyses of the genes located in these regions. Unlike earlier work examining fewer, less well localized hotspots (*Yvert et al., 2003*), we found a strong enrichment for transcription factors, a class of genes with obvious relevance to gene regulation. Guided by this enrichment, we noticed that almost all tran-scription factor genes with predicted damaging variants in this cross appear to produce a *trans*-act-ing hotspot, with the exception of two genes in which the damaging variants may affect gene function only weakly if at all. This result clearly illustrates the importance of variation in transcription factors in shaping gene expression in trans. At the same time, many *trans*-acting hotspots are caused by variation in genes other than transcription factors. These hotspots illustrate the importance of indirect *trans* effects that alter cellular networks and physiology, which in turn results in gene expres-sion change.

In an unbiased search, we revealed hundreds of non-additive genetic interactions that influence mRNA levels. Many of these eQTL-eQTL interactions were between eQTLs at two different hotspot regions, or interactions between an eQTL at a hotspot and a local eQTL. Hotspots may play such

prominent roles in epistatic interactions because their wide-reaching effects effectively alter the global state of the cell. The effects of many other variants may be larger in some cellular states than others, similar to what is seen when yeast are grown in different environments (*Lewis et al., 2014*; *Smith and Kruglyak, 2008*).

Previous searches for epistatic interactions on gene expression levels in this cross were targeted to genes and loci chosen based on prior additive scans (*Brem et al., 2005*; *Storey et al., 2005*). Our unbiased search confirmed the validity of this strategy. The vast majority of epistatic markers we detected also had additive effects, and there was little evidence for interactions without any additive effects. Conversely, many additive eQTLs, including several major additive hotspots, were involved in few if any epistatic interactions (*Figure 7A*). Thus, epistatic interactions cannot be simply assumed to exist for all additive loci, and the most fruitful strategy to detect epistasis is to test loci with additive effects for interactions with other loci.

In our cross between two yeast strains, we identified over one hundred hotspot regions, and found evidence for additional, weaker hotspots. Studies in many multicellular organisms, including plants (*Cubillos et al., 2012*; *Fu et al., 2009*; *West et al., 2007*; *Zhang et al., 2011*), nematodes (Li et al., 2006; *Rockman et al., 2010*), mice (*Hasin-Brumshtein et al., 2016*; *Kelly et al., 2012*; *Orozco et al., 2012*) and rats (*Heinig et al., 2010*; *Hubner et al., 2005*; *Kaisaki et al., 2016*) have also observed *trans*-eQTL hotspots, showing that this phenomenon is not restricted to yeast. Compared to our yeast cross, human populations harbor orders of magnitudes more variants, and the human genome has many more genes and more extensive and complex regulatory regions that offer a large target space for regulatory variation. We would expect these characteristics to result in human *trans*-eQTL architectures even more complex than that we have uncovered here in yeast.

Recent studies have made progress in identifying *trans*-eQTLs in humans. Many of the human *trans*-eQTLs discovered to date tend to influence the expression of multiple genes (*Battle et al., 2014*; *Brynedal et al., 2017*; *Fehrmann et al., 2011*; *Grundberg et al., 2012*; *Heinig et al., 2010*; *Lee et al., 2014*; *Small et al., 2011*; *Wright et al., 2014*; *Yao et al., 2017*). Specifically, work in blood and blood-derived cells from up to ~5000 individuals (*Battle et al., 2014*; *Fairfax et al., 2012*; *Fehrmann et al., 2011*; *Pierce et al., 2014*; *Westra et al., 2013*; *Wright et al., 2014*; *Yao et al., 2017*) revealed SNPs that were associated with multiple genes in trans, including the HLA locus (*Fehrmann et al., 2011*), LZY in monocytes, and KLF4 in B-cells (*Fairfax et al., 2012*). Multiple genes linking to single SNPs were also found in human adipose tissue (e.g.in the KLF14 transcription factor gene (*Aguet et al., 2017*; *Hore et al., 2016*; *Small et al., 2011*, *2018*), skin and lymphoblastoid cell lines (*Grundberg et al., 2012*), and other tissues (*Aguet et al., 2017*). In immune cells, condition-specific hotspots at known regulators of the immune response were seen upon stimulation, and these altered expression of specific immune response pathways (*Fairfax et al., 2014*; *Lee et al., 2014*). Shared, weak effects across transcripts (*Brynedal et al., 2017*; *Hore et al., 2016*) indicated that 'hundreds of *trans*-eQTLs each affect hundreds of transcripts' (*Brynedal et al., 2017*). Together, these observations show that *trans*-eQTL hotspots exist in humans.

Whether *trans*-eQTL hotspots are as predominant in humans as they are in yeast (where 90% of our *trans*-eQTLs mapped to a hotspot) remains an open question. Human *trans*-eQTLs could be distributed more uniformly, such that hotspots make up a smaller fraction of all *trans*-eQTLs, leaving room for more gene-specific *trans*-eQTLs. There could also be more hotspots, each affecting fewer genes than the strongest hotspots in our cross. While it is difficult to extrapolate from the current, still underpowered human studies, it may be informative to consider that the initial eQTL searches in our yeast cross only found eight hotspots, each affecting at most dozens of genes (*Brem et al., 2002*). The observation that hotspots are the main source of *trans* variation required the higher statistical power of the present dataset. It remains to be seen if human *trans*-eQTL discovery in larger samples will follow a similar trajectory.

The recently proposed omnigenic model (*Boyle et al., 2017*) for the genetic basis of complex trait variation posits that gene regulatory networks are sufficiently densely connected that the change in expression of any one gene, caused by a local eQTL, will 'percolate' through the network and alter the expression of all other genes that are expressed in a given cell type. The hotspot loci we described here offer evidence that some regulatory variants can indeed have widespread effects on the transcriptome, in some cases altering the expression of the majority of genes in the genome through precisely the combination of strong direct effects on 'core' genes in specific pathways and

weak indirect effects on other 'peripheral' genes envisioned in the omnigenic model. Hundreds of hotspots may exist in the human genome (*Brynedal et al., 2017*), providing a rich substrate through which regulatory variation may influence complex traits.

On the other hand, although we detected local eQTLs for most genes in our cross, the majority of these had no detectable *trans* effects on the expression of other genes, within the limits of our statistical power. Given that our study had sufficient power to detect weak indirect effects of *trans*-eQTL hotspots, we believe that most local eQTLs indeed have no meaningful downstream consequences for gene expression. By extension, such local eQTLs may be unlikely to contribute to variation in complex traits. Consistent with this conclusion, modest expression changes for dozens of yeast genes have been found to result in minimal fitness effects (*Duveau et al., 2017*; *Keren et al., 2016*). These results argue against the simplest form of the omnigenic model, in which a variant that changes the expression of any one gene has meaningful effects on every other gene. Instead, we observed that *trans*-eQTLs effects preferentially arise from variation in certain classes of genes. Given the crucial importance of regulatory variation for many complex traits (*Albert and Kruglyak, 2015*), the organismal consequences of expression changes caused by different types of eQTLs remain a key area for further research.

## Materials and methods

Unless otherwise specified, all computational analyses were performed in R. Analysis code is available at https://github.com/joshsbloom/eQTL_BYxRM (*Bloom and Albert, 2018*; copy archived at https://github.com/elifesciences-publications/eQTL_BYxRM). Supplementary Data files are also available at https://figshare.com/s/83bddc1ddf3f97108ad4.

### Yeast growth

We used 1012 meiotic segregants previously generated (*Bloom et al., 2013*) from a cross between the prototrophic yeast laboratory strain BY (*MATa*; derived from a cross between BY4716 and BY4700) and the prototrophic vineyard strain RM (*MATα hoΔ::hphMX4 flo8Δ::natMX4 AMN1-BY*; derived from RM11-1a). The segregants were grouped according to their previously measured (*Bloom et al., 2013*) endpoint colony radius on YNB agar plates into groups of 96. The strains in each group were rearranged from existing stock plates into a total of 13 96-well plates in YNB medium, grown to saturation, and frozen as glycerol stocks for later growth. Within each group of 96, strain locations in the 96-well plate were selected at random. Culture and liquid handling was performed on a BioMek FXP instrument or with multichannel pipettes in 96-well format.

Our strategy of batching segregants according to their growth on YNB ensures that each 96-well plate contains segregants that grow at comparable rates. This facilitates growing all segregants on a plate such that they reach a similar optical density at 600 nm (OD) at the same time. Our batching strategy produces experimental batches that are correlated with growth rates. Because we statistically removed variation among experimental batches prior to eQTL mapping (see below), this design reduces our ability to compare variation in growth rates with variation in gene expression. We deemed this an acceptable trade-off because it considerably simplified handling >1000 samples in a systematic fashion. We processed the batches in a randomized order with respect to their growth rate to avoid confounding processing date with faster or slower growth.

We used the rearranged stock plates to inoculate growth cultures in 1 ml YNB medium (recipe for 1 L: 6.7 g yeast nitrogen base with ammonium sulfate and without amino acids; 900 ml $H_2O$; autoclave; add 100 mL of separately autoclaved 20% glucose solution) in 2 mL deep well plates sealed with Breathe-Easy membranes (Sigma Aldrich), and grew the cultures to saturation on Eppendorf MixMate instruments situated in a 30°C incubator and set to 1100 rounds per minute (rpm). We set the saturated cultures back to OD = 0.05 in 1 mL YNB in a fresh deep well plate and continued growth at 30°C. We monitored OD during growth by splitting out 100 μL of culture every other hour, measuring OD on a Synergy two plate reader (BioTek) and returning the 100 μL used for measuring OD to the deep-well culture plate. We increased the frequency of measurements as cultures approached OD = 0.4.

Once average OD in the plate reached 0.4, we transferred the cultures to sterile Norgen nylon filter plates (#40008) situated on a vacuum manifold. We applied vacuum to remove all growth medium, sealed with aluminum foil seals, and flash froze the entire plate in liquid $N_2$. The frozen

plates were placed on a standard 96-well plate to protect their bottom, wrapped with parafilm, and stored at −80°C until RNA extraction. Note that this procedure provided us with OD measurements up to the exact time point at which cells were harvested.

## RNA extraction

We used Dynabeads mRNA DIRECT kits (Ambion/Thermo Fisher) to directly isolate mRNA from cell lysates. To perform the RNA extractions on the BioMek robot, we prepared excess lysis/binding and Wash buffers that permitted the use liquid reservoirs with volumes that exceed that provided in the kits. These buffers were prepared as specified in the Dynabeads kit protocol:

Lysis/Binding Buffer:
100 mM Tris-HCl, pH 7.5
500 mM LiCl
10 mM EDTA, pH 8
1% LiDS
5 mM dithiothreitol (DTT)
Washing Buffer A:
10 mM Tris-HCl, pH 7.5
0.15 M LiCl
1 mM EDTA
0.1% LiDS
Washing Buffer B:
10 mM Tris-HCl, pH 7.5
0.15 M LiCl
1 mM EDTA

We filled the wells of an Axygen 1.1 mL plate (P-DW-11-C-S) with about 250 µl acid washed 425–600 µm beads (Sigma G8722). We added 700 µL lysis buffer to our frozen cell plates, pipetted up and down to resuspend the cells, and applied them to the glass beads in the Axygen plate. The Axygen plate was tightly sealed with an Axymat rubber plate seal (AM-2ML-RD-S), and ground for 10 cycles on a plate-based mini bead beater (Biospec). Each cycle consisted of 1 min beating followed by 1 min on ice.

We centrifuged the plate for 4 min at 3000 rpm to separate glass beads and cell debris from the lysate. We pipetted two aliquots of 200 µL of lysate supernatant into two 96-well PCR plates for a total of 400 µL lysate. These plates were sealed, and the RNA melted for 2 min at 65°C in a thermocycler. We implemented a BioMek-assisted procedure to perform the Dynabead protocol with two mRNA enrichment steps. We did not quantify the resulting 11 µL of mRNA and simply used the entire mRNA for reverse transcription and sequencing library preparation. While piloting this procedure, we obtained typical yields of ~30 ng / µL and excellent RNA quality as judged by visualization on 1.1% agarose gels stained with ethidium bromide. Ribosomal RNA bands were clearly visible in crude lysate, less visible after the first mRNA enrichment, and absent after the second mRNA enrichment step. After the second mRNA enrichment, mRNA was clearly visible on the gel, with no visible RNA degradation.

## RNA sequencing library construction and sequencing

We performed reverse transcription and sequencing library preparation using the Kapa Stranded mRNA-Seq Kit (KK8420/21). This kit usually begins by enriching mRNA from total RNA. Because we had already performed mRNA enrichment, we used our entire mRNA as input and began at the RNA fragmentation step by adding 11 µL of 'KAPA fragment, prime and elute buffer' to our 11 µL of mRNA. RNA fragmentation was performed on a thermocycler for 6 min at 94°C.

The remaining procedure was performed as specified in the Kapa kit manual. Briefly, the fragmented RNA is randomly primed and used for first strand cDNA synthesis, second strand synthesis and marking with dUTP, A-tailing of the double-stranded cDNA, adapter ligation, and PCR for 12 cycles. The dUTP marked second strand is not amplified in PCR, resulting in strand-specific libraries. We used custom designed Truseq-compatible indexing adapters (IDT) to allow multiplexing all 96 samples per batch. Prior to use, the two types of Truseq adapters were annealed (2 min at 97°C; 72

steps of 1 min at 1°C decreasing temperature; 5 min at 25°C) to generate forked adapters that can be ligated to the A-tailed cDNA. We did not pool samples between batches.

Sequencing libraries were quantified by combining 1 µL of library with 100 µL of Qubit High Sensitivity dsDNA reagent in 96-well plates with black bottom and wells, and reading fluorescence (excitation 485 nm, emission 528 nm) on the Synergy two plate reader. We calculated library concentrations by comparing to a standard series obtained by diluting the standard solutions included in the Qubit quantification kit. Standards were measured in triplicate on each library plate. We pooled the libraries in each group to equal molarity and used qPCR (KAPA Biosystems #KK4854) on the pool to obtain the molarity for loading on the sequencer. Gel extraction was not necessary because the RNA fragmentation and bead clean-up that are part of the Kapa protocol resulted in library fragments of the desired size of 200–400 bp.

Sequencing was performed for 100 bp single end on Illumina HiSeq 2500 instruments at the UCLA BSRC sequencing core for two lanes per batch, for 26 total lanes. On average, we obtained approximately 3 million reads per sample. Sequencing reads are available in SRA under the accession codes listed in the data availability statement.

## Sequence processing and gene expression quantitation

Adapter sequences were trimmed using trimmomatic (*Bolger et al., 2014*). Reads were pseudoaligned to the 6713 annotated yeast ORF coding sequences from Ensembl build R64-1-1 using kallisto v.43.0 (*Bray et al., 2016*). Kallisto was run in strand-specific mode with parameters –l 150 and –s 8. For each transcript, we computed transcripts per million reads (TPM) as a measure of expression and used $\log2(TPM + 0.5)$ for downstream analysis. Segregants with fewer than one million reads were removed from downstream analysis, and 1012 segregants passed this filter. We removed 993 invariant transcripts with identical expression across all segregants or with $\log2(TPM + 0.5)$ less than 1 in 50% or more of the segregants. Our final dataset included 5720 transcripts, which were used for downstream analyses (*Source data 1*). These transcripts cover 5506 of 5971 open reading frames annotated as 'verified' or 'uncharacterized' in the yeast genome (*Cherry et al., 2012*).

## Growth rate covariate

Unless otherwise specified, all remaining analyses were conducted in R (www.r-project.org). Based on the OD measurements collected during growth prior to harvesting, growth rates were calculated for each segregant using the R package grofit and the function gcFitSpline (*Kahm et al., 2010*). The difference between the maximum and minimum OD was recorded for each culture and used as a covariate for downstream analysis (*Source data 2*).

## Sequence variants

Our BY and RM parent strains had earlier been sequenced to very high depth (>200 fold coverage of the genome), and GATK (*McKenna et al., 2010*) used to identify 48,254 sequence variants between them. These variants (irrespective of whether or not they are part of our marker map) were screened for potential functional impact using the Ensembl Variant Effect Predictor (*McLaren et al., 2016*).

The segregant genotyping is described in (*Bloom et al., 2013*) and (*Bloom et al., 2015*). The 1012 segregants used for this study were genotyped at 42,052 highly reliable markers, which are a subset of the total 48,254 sequence differences between BY and RM. Sets of markers that were in perfect linkage disequilibrium (i.e. markers never separated by recombination) among the 1012 segregants were collapsed to one marker. Our final linkage map comprised 11,530 unique markers (*Source data 3*).

## Heritability

A variance component model was used to estimate additive heritability. First, gene expression measurements were corrected for batch covariates and the growth measurement covariate described above using a linear model for each gene

$P=D\mathbf{G} + R$

where $P$ is a vector of $\log_2(TPM + 0.5)$ measurements for $n$ segregants for that gene. $D$ is a vector of estimated fixed effect coefficients for technical covariates. $\mathbf{G}$ is a matrix of $n$ total segregants by

*m* technical covariates. Technical covariates included experimental batch and the growth rate covariate described above. The vector of residuals is denoted as *R*. *R* contains expression phenotypes corrected for batch effect and growth covariate.

We fit the variance component model

$R = a + e$

where *a* is the vector of additive genetic effects, and the residual error is denoted by *e*. The distributions of these effects are assumed to be normal with mean zero and variance–covariance as follows:

$a \sim N(0, \sigma^2_A A)$; $e \sim N(0, \sigma^2_{EV} I)$

Here, **A** is the additive relatedness matrix – the fraction of the genome shared between pairs of segregants. **A** was calculated using the 'A.mat' function in the rrBLUP R package (*Endelman, 2011*). $\sigma^2_A$ is the additive genetic variance captured by markers. $\sigma^2_{EV}$ is the error variance and *I* is the identity matrix. Additive heritability was estimated using custom code adapted from Kang et al (*Kang et al., 2008*).

Although our heritability estimates are lower bounds due to counting noise in the number of sequencing reads per gene, downsampling of reads suggested that additional sequencing would increase heritability for most genes by at most a few percent (*Figure 1—figure supplement 1*).

Additionally, we fit a model to estimate the relative contribution of pairwise interactions with

$R = a + i + e$

where $i \sim N(0, \sigma^2_{AA}(A \degree A))$ and **A°A** is the Hadamard (entry-wise) product of **A**, which can be interpreted as the fraction of pairs of markers shared between pairs of segregants. $\sigma^2_{AA}$ is the interaction genetic variance captured by all pairwise combinations of markers. The other terms are the same as in the additive-only model. The result of ~1/10 as much variance arising from interactions relative to additive loci is based on the ratio of the average of the **A°A** term across genes to the average of the **A** term across genes. When we instead calculated this variance ratio for each gene, we found that the mean across genes was greatly inflated by a few extreme outliers, while the median was very low (less than 1/100) because almost half of the genes had an estimate of zero for the **A°A** term.

## Gene annotations and features

Gene positions were extracted from Ensembl (*Yates et al., 2016*) (www.ensembl.org) build 83. Various analyses throughout the paper made use of a range of gene-specific features, factors and covariates: (1) Total variance in expression was calculated as the sum of the additive and residual variance components obtained in our heritability estimates. (2) Expression level was calculated as the mean log2(TPM) across segregants, (3) Gene essentiality was coded as a binary factor and obtained from SGD (*Cherry et al., 2012*) (www.yeastgenome.org) by searching for genes whose SGD deletion phenotype contained the term 'inviable'. (4) dN/dS values were obtained from Supplementary Table S4 in (*Wall et al., 2005*). (5) The number of protein-protein interactions was obtained from SGD by downloading all 'physical' interactions between genes and counting their number per gene. (6) Synthetic genetic interactions were extracted from data from *Costanzo et al. (2016)* which provides genetic interaction data for pairwise gene deletions or disruptions between nearly all essential (E) and nonessential (N) genes (*Costanzo et al., 2016*). Specifically, we downloaded the 'NxN', 'NxE', and 'ExE' raw genetic interaction datasets from http://thecellmap.org/costanzo2016/, combined them into one table, and extracted the lowest interaction p-value for each gene pair. We restricted this set using the 'strict' definition from (*Costanzo et al., 2016*) and kept only pairs with interaction p-value<0.05 and interaction strength (epsilon) >0.16 or<−0.12. For each gene, we counted how many genes showed a genetic interaction at these thresholds and used this as our measure of synthetic genetic interactions. Using the 'lenient' or 'intermediate' definitions did not alter our conclusions. (7) We defined whether or not a gene is a transcription factor by downloading from SGD all genes annotated to the GO term GO:0003700 'transcription_factor_activity_sequencespecific_DNA_binding' and its child GO terms. (8) As a proxy for deep evolutionary conservation, we extracted from Ensembl biomart whether or not a gene has a human homolog.

Gene ontology (GO) associations for each gene were downloaded from the Gene Ontology Consortium (geneontology.org) on February 16, 2016. We used paralogy information downloaded from the yeast gene order browser (*Byrne and Wolfe, 2005*) (http://ygob.ucd.ie/).

## Characteristics of genes with high or low heritability

We tested for gene features associated with the degree of heritability by multiple linear regression. This regression modeled heritability as the dependent variable and the various gene features as predictor variables. We used the 'summary' and 'lm' functions, and the 'car' package in R to perform Type III sum-of-squares ANOVA. This analysis tests for the influence of each feature by dropping it from a full model that includes all other terms, and asking whether this results in a significantly worse fit as judged by F-statistics. The analysis controls for correlations among predictor variables and reports marginal associations only if they are significant over all other terms. We did not include interaction terms among predictor variables.

## Gene ontology enrichment analyses

We tested for GO enrichments using the R package topGO (*Alexa et al., 2006*). For analyses in which genes were classified as 'interesting' or not (e.g. whether a gene has heritability ≥90%, or whether it is located in a hotspot), we used the Fisher test for enrichment. When using a quantitative gene score as the measure of interest (e.g. the heritability), we used the one-sided t.test implemented in topGO. We used the 'classic' scoring method (*Alexa et al., 2006*), that is we did not adjust the enrichments for significance of child GO terms.

## eQTL mapping

We (*Bloom et al., 2015*) and others (*Yang et al., 2014*; *Zeng, 1994*) have previously noticed that power and precision of QTL mapping on a given chromosome can be increased by controlling for genetic contributions that arise from the other chromosomes in the genome. Our eQTL mapping strategy controls for genomic background in two ways. For each gene, we identified large genetic effects segregating on other chromosomes and included them as covariates while mapping on a given chromosome. We also corrected for any additional polygenic additive background signal on other chromosomes. Then, for each gene we used a forward stepwise procedure to map eQTLs with a false discovery rate procedure. Below we describe our algorithm in greater detail. Throughout, we use the terms 'eQTL' and 'linkage' interchangeably.

### Identification of large background genetic effects

In the process of eQTL mapping on a given chromosome, we wanted to control for the genetic contributions from the remainder of the genome. Although this background control is sometimes done using a polygenic model with a random effect that captures overall relatedness across the genome (*Yang et al., 2014*) (see below), large individual eQTL effects may not be adequately accounted for by one genome-wide relatedness matrix. Therefore, we performed the following procedure to identify large genetic effects. Our goal at this stage was not to formally identify these large effects as eQTLs, but to perform a simple scan for large effects that can be included as covariates to control for their effects while mapping on a given chromosome. As our algorithm progresses along each chromosome, these large background effects will eventually be detected as formal eQTLs.

1. Gene expression measurements were corrected for batch covariates and the growth covariate using a linear model

$$P = DG + R$$

where $P$ is log2(TPM + 0.5), $D$ is the vector of estimated fixed effect coefficients for technical covariates (batch and growth), and $G$ is a matrix of $n$ total segregants by $m$ technical covariates as described in the section on Heritability above. The residual vector is denoted as $R$ and contains the corrected expression measurements.

We sought to identify a set of markers linked to large-to-moderate effect QTLs. For each gene and for each chromosome, we calculated logarithm of the odds (LOD) scores between corrected expression levels $R$ and the marker genotypes on that chromosome as

$$-n(\ln(1-r^2)/2\ln(10))$$

where $n$ is the number of segregants with genotypes and phenotypes, and $r$ is the Pearson correlation coefficient between segregant genotypes and $R$. The marker with the largest LOD per chromosome was added to a matrix, $Z$, if it had LOD > 3.5.

For each gene we calculated

$$P = DG + CZ + R$$

where $C$ is a vector of genotype effects. This step controls for technical covariates as above, but additionally controls for large effects included in $\mathbf{Z}$. It results in a new vector of residual expression levels $R$. We repeated steps 2 and 3 twice with this new $R$, appending additional markers to $\mathbf{Z}$ for each gene and each chromosome if they passed the threshold of LOD > 3.5. The goal of this repeated search for large effect eQTLs was to control for large effect loci that were detected only after expression values were corrected for the effects of previously identified large effect loci. For example, a given chromosome may harbor several large effect QTLs, and repeated runs of steps 2 and 3 ensure that such loci are captured. At the end of this procedure, we had a matrix $\mathbf{Z}$ of up to three markers per chromosome per gene that were linked to large-to-moderate effect loci.

## Correcting gene expression measurements for large background effects and additive polygenic background for all chromosomes except the chromosome of interest

For each chromosome of interest and for each gene expression trait we calculated

$P = D\mathbf{G} + C_L\mathbf{Z_L} + a_L + R$

$C_L$ and $\mathbf{Z_L}$ are the background eQTL effects identified from the procedure above that are not located on the chromosome of interest. $a_L \sim N(0, \sigma^2_{aL}\mathbf{A_L})$ $\sigma^2_{aL}$ is the additive genetic variance from all chromosomes excluding the chromosome of interest. $\mathbf{A_L}$ was calculated using the 'A.mat' function in the rrBLUP package using a genetic relatedness matrix that excludes markers from the chromosome of interest. The goal of this step was to obtain expression phenotypes $R$ that can be used to scan for eQTLs on a given chromosome by correcting for sources of variation that do not arise from that chromosome: batch and growth effects, large effects on other chromosomes, and a polygenic term accounting for any additional genetic contributions arising from other chromosomes.

### Mapping additive eQTLs

We mapped additive eQTLs using a forward stepwise procedure. For each chromosome and for each gene we tested for linkage at each maker on the given chromosome with residual expression values $R$ (calculated above) using the formula in Step 2 of 'Identification of large background genetic effects'. We recorded the location and LOD score of the marker with the highest LOD score. To decide if this marker should be included as a QTL in the model, we used a permutation-based FDR criterion of 5%.

FDR was calculated as the ratio of the number of genes expected by chance to show a maximum LOD score greater than a particular LOD threshold vs. the number of genes observed in the real data with a maximum LOD score greater than that threshold, for a series of LOD thresholds ranging from 1.5 to 9 with 151 equal-sized steps of 0.05. The number of genes expected by chance was calculated by permuting $R$ relative to segregant genotypes, calculating LOD scores for all genes across the chromosome and recording the maximum LOD score for each gene. In each run of the permutations, the permutation ordering was the same across all genes. We repeated this permutation procedure 1000 times. Then, for each of the 151 LOD thresholds, we calculated the average number of genes with maximum LOD greater than the given threshold across the 1000 permutations. We found the lowest of the 151 LOD thresholds at which the ratio of the number of expected genes with an eQTL of at least this threshold to the number of observed genes of at least this threshold was <5%. This LOD threshold was used as a criterion for declaring a given QTL in the real data as significant.

For all genes with a significant linkage (FDR < 5%), the peak marker was added to the linear model for that gene as $X$:

$R = QX + e$

Genes without a significant linkage were excluded from additional testing on that chromosome. For the set of genes with a significant linkage, we repeated the procedure above by replacing $R$ with $e$. The procedure was repeated for each chromosome until no genes had additional significant linkages.

## Cross-validation

The amount of additive variance explained by detected eQTLs was estimated using cross-validation. Segregants were grouped based on the batches used for RNA and library preparation. Each batch of segregants was left out of the procedure one at a time. The eQTL mapping procedure was

performed for all the other batches. For the QTL markers detected in this training set and with effects estimated in the training set, the amount of variance explained by the joint model of the set of significant QTL markers was estimated in the held out batch.

### eQTL confidence intervals

eQTL confidence intervals were calculated as 1.5 LOD drops. We extended the eQTL location confidence intervals to include all markers in perfect LD with the markers used in eQTL detection (marker correlation = 1).

### Hotspot identification

We devised an algorithm with the goal of identifying a set of eQTL hotspots by combining information across genetically correlated transcripts and, most importantly, using co-localizing *trans*-eQTLs to better narrow hotspot confidence intervals. The algorithm has three major steps. First, we control for unmodeled factors affecting gene expression that may obscure hotspot detection and localization. Second, we use a multivariate statistic to identify eQTL hotspots. Finally, we use a bootstrap procedure to delineate confidence intervals for hotspot location. We describe the steps in greater detail below.

#### Identification of unmodeled factors affecting gene expression

For each gene with at least one statistically significant eQTL we fit a linear model

$$P = D\boldsymbol{G} + Q\boldsymbol{X} + a + R_e$$

where $P$ is log2(TPM +0.5), $D$ is the vector of estimated fixed effect coefficients for technical covariates, and $\boldsymbol{G}$ is a matrix of $n$ total segregants by $m$ technical covariates as described in the section on Heritability. $\boldsymbol{X}$ is a matrix of the statistically significant additive QTL peak markers for the gene, $Q$ is a column vector of QTL effects, and $a$ is random effect for the contribution of polygenic background to additive variance, where $a \sim N(0, \sigma^2_a \boldsymbol{A})$, $\sigma^2_a$ is the additive genetic variance from all markers and $\boldsymbol{A}$ is constructed as described above in the section on calculating heritability. $R_e$ is the residual expression level after the contributions of all additive influences on gene expression variation in the model have been removed.

For each gene, $R_e$ was scaled to have mean 0 and variance 1. $R_e$ for each gene was concatenated to form the columns of the matrix $\boldsymbol{R}$.

We calculated the singular value decomposition (SVD) of $\boldsymbol{R}$. We also calculated the SVD after each column of $\boldsymbol{R}$ was individually permuted. We visually inspected a Scree plot for both decompositions and observed that the top 20 eigenvectors explained more variance than expected by chance.

The top 20 eigenvectors were appended to the matrix $\boldsymbol{G}$ of covariates. These eigenvectors capture systematic expression variation that is shared across genes but that does not arise from any known technical covariate, identifiable eQTLs, or additive genomic background as captured by our markers. These components presumably reflect undetected sources of variation from unmodeled experimental factors or non-additive genetic factors shared across genes. Their inclusion as covariates in $\boldsymbol{G}$ below follows from the same logic that motivates SVA (*Leek and Storey, 2007*); we want to account for unmodeled factors that are contributing to gene expression variation to increase power to map and resolve eQTL hotspots.

We note that because we cannot explicitly model all possible expected *trans* effects per gene first as fixed effects, it is theoretically possible that small effects not captured by the QTL model or additive polygenic background per gene could contribute to the matrix $\boldsymbol{R}$ and the top 20 eigenvectors. Regressing out genetic effects captured by these eigenvectors could then induce spurious *trans*-associations (*Dahl et al., 2017*).

#### Correcting gene expression for technical covariates, detected additive eQTLs on other chromosomes, and unmodeled factors

For each chromosome, we extracted all genes that have a significant linkage to that chromosome but do not physically reside on that chromosome (i.e. all genes that had a statistically significant *trans*-eQTL on the given chromosome) and fit the linear model

$$P = D\boldsymbol{G} + Q_L\boldsymbol{X}_L + R_e$$

This model is similar to that in Step 1, 'identification of unmodeled factors affecting gene expression', except that the matrix $G$ is extended to include the 20 eigenvectors corresponding to unmodeled factors, and $X_L$ only contains statistically significant additive eQTLs that do not reside on the chromosome of interest. Thus, $R_e$ is corrected for all influences on expression that are not relevant to mapping hotspots on the given chromosome, but retains all signal that does arise from the chromosome of interest.

For each gene, $R_e$ was scaled to have mean 0 and variance 1. $R_e$ for each gene was concatenated to form the columns of the matrix $R$.

## Dimensionality reduction of R

Our algorithm for fine-mapping eQTL hotspots makes use of a multivariate statistic (see section 'multitrait mapping to localize eQTL hotspots'). We observed that on many chromosomes, the number of genes with *trans* linkages exceeded our total sample size. Because the multivariate statistic is not defined in this case, we reduced the dimensionality of $R$ using SVD. The top $m$ eigenvectors with corresponding eigenvalues greater than those observed from an SVD on permuted data were retained as matrix $L$. The entries of $L$ can be interpreted as weighted linear combinations of the expression levels from individual genes, from which all sources of variation that do not arise from the chromosomes of interest have been eliminated. These combinations capture the majority of additive genetic influences on the given chromosome that are shared across genes (i.e. the effects of hotspots on multiple genes), and serve as the input to our hotspot detection algorithm.

## Multitrait mapping to localize eQTL hotspots

We computed

$L = u +$ BX$+E$

where $u$ is a vector of means for each of the columns of $L$, $B$ is a vector of coefficients for the effects of $X$ on each of the columns of $L$, $X$ is the genotype vector for each marker on the target chromosome (the model is fit one marker at a time), and $E$ is a matrix of residuals. Next, we calculated the residual sum of squares matrix $RSS$ as $E'E$ and LOD scores as $(n/2)\log_{10}(|RSS_0|/|RSS|)$, where $|RSS|$ denotes the determinant of the $RSS$ matrix and $RSS_0$ is the residual sum of squares matrix for the null model with no QTL effect (*Knott and Haley, 2000*; *Tian et al., 2016*). This LOD score reflects linkage of genetic markers with the joint expression of the eigenvectors in $L$. To distinguish significant linkages affecting multiple genes from eQTLs for individual genes, we refer to them as jointQTLs (jQTLs) in the remainder of this method section.

We fit the equation above 100 times with permutations of phenotype to genotype for matrix $L$. The 99% quantile of the maximum observed LOD score per permutation was used to decide whether the maximum multivariate LOD was significant. If it was significant, then the effect of that marker was subtracted from $L$ and the procedure repeated until no more jQTLs were detected.

## Refining jQTL locations and test for hotspots that are better modeled as two neighboring hotspots

We sought to better refine the location of the jQTLs given all other jQTLs on that chromosome (*Zeng et al., 1999*). We collected the statistically significant peak markers for the jQTLs identified above. Each peak marker was dropped from a joint model one at a time and the peak position was recomputed. Peaks that moved by more than 75 kb at this step were removed from the model. For all peaks with LOD >200 we tested the possibility that the multivariate peak was a 'ghost' jQTL, that is a jQTL that appears to be localized between two or more true jQTLs that are very closely linked and that influence a similar set of genes. We fit a model where the one significant jQTL was replaced by a two-jQTLs model, with the added criterion that the two jQTLs could not be within 10 markers on either side of the original peak. Permutations for this two-locus model were performed as in 'multitrait mapping to localize eQTL hotspots', with the relevant test and null statistics being the difference between the fit of the best two-locus model and the best one-locus model. Single jQTL peaks were replaced with the best two-locus model if the observed LOD was more than 10 greater than the 99% quantile LOD for the permuted data. Through the combination of the steps above, this procedure identified significant jQTLs that correspond to hotspots that influence the expression of multiple genes.

## Bootstrap resampling to identify confidence intervals for jQTL location

For each detected jQTL, segregants were sampled with replacement 1000 times. For each sample, we refit

$$L_{si} = u + B_s X_s + B_{si} X_{si} + E$$

where $s$ indicates a bootstrap resample of the rows of a matrix. $X_s$ contained the other significant jQTL peak markers from the interrogated chromosome and $X_{Si}$ is the genotype vector for each marker on the target chromosome within a window of 80 markers centered on the observed QTL peak being interrogated, with the effect of each marker fit one at a time. LOD scores were computed as in Step 1 of 'multitrait mapping to localize eQTL hotspots' and the position of the marker with the maximum LOD score was retained for each bootstrap resampling.

For all further analyses, we defined hotspot location confidence intervals as the central 95% confidence interval of bootstrap peaks. These intervals were extended to include all markers that are in perfect linkage disequilibrium with the markers at the ends of the confidence intervals.

## Local eQTLs

To classify an eQTL as local, we required its location confidence interval to overlap the position of the gene. We used gene locations expanded by 1000 bp upstream and 200 bp downstream to account for regulatory variants that may be located in the promoter or the 3'UTR. We initially classified 2969 eQTLs as local. These 2969 eQTLs affected 2884 genes. Closer inspection revealed that these multiple 'local' eQTLs per gene often involved one eQTL with a peak very close to the gene and other, more distant eQTLs. These more distant eQTLs probably reflect *trans*-eQTLs on the same chromosome as their target gene with location confidence intervals broad enough that they happened to overlap the target gene. For our ASE comparisons below, we only used the local eQTLs that were located closest to a given gene.

## Allele-specific expression analyses and comparison to local eQTLs

We used ASE data from two sources. The first source is the mRNA data from Supplementary Data S2 from (*Albert et al., 2014a*). The second source is previously unpublished data generated by Dr. Noorossadat Torabi in the Kruglyak laboratory (available in SRA under accession code SRP149494). Both datasets performed mRNA sequencing on a BY/RM diploid hybrid strain. Reads from Albert et al. were ~30 bp in length to match ribosome profiling data presented in that paper, while reads from Torabi et al. were 100 bp. In contrast to (*Albert et al., 2014a*), the Torabi data was not strand-specific.

We processed the Torabi data exactly as described in (*Albert et al., 2014a*). Briefly, reads were aligned to both the BY reference genome and the RM reference genome provided by the Broad Institute (https://www.broadinstitute.org/fungal-genome-initiative/saccharomyces-cerevisiae-rm11-1a-genome-project). We retained reads that mapped uniquely and without mismatch. We considered reads mapping to a set of coding SNPs carefully curated to only contain SNPs with good mapping characteristics (*Albert et al., 2014a*) and counted the number of reads arising from each allele. Counts for multiple SNPs per gene were summed. We performed hypergeometric downsampling to account for a small difference (0.5%) in total read counts mapping to BY vs. RM alleles. Genes with fewer than 20 reads were discarded. Significance of ASE was gauged using a binomial test, and p-values Bonferroni-corrected for multiple testing. Effect sizes are expressed as log2-transformed fold changes, which in turn are calculated as the RM allele count divided by the BY allele count for each gene.

The Torabi and Albert dataset are independent replicates of a BY/RM hybrid. We can use this fact to gain confidence in significance calls (we keep track of whether a gene was determined to have significant ASE in one or both of the datasets) and the magnitude of ASE (for each gene, we use the mean log2 fold change of the two datasets). Across the two datasets, we had ASE data available for 3340 genes.

To compare effect sizes between eQTLs and ASE, we used log2-transformed eQTL fold changes re-computed on un-scaled expression data. ASE fold changes were computed as the log2 of the ratio of RM to BY allele counts. Comparisons of effect size across genes were performed using standardized major axis (SMA) analysis (*Warton et al., 2012*). To correlate ASE and eQTL effect sizes to the number of sequence variants upstream of each gene, we defined the upstream interval as the

sequence upstream of the start codon up to the neighboring gene for a maximum of 1,000 bp. We acknowledge that the length of sequence considered is therefore different for different genes but believe that this is a reasonable approximation of the regulatory upstream regions in yeast (*Lin et al., 2010*). Sequence variants had been obtained from short-read sequencing of the BY and RM strains used in this study (*Bloom et al., 2013*).

All data necessary to reproduce the ASE analyses and eQTL comparisons is available in *Source data 7*.

## Power analyses for ASE data

To gauge the statistical power to detect ASE in the two available ASE datasets, we performed simulations. We focused on two variables: the effect size (i.e. fold change) and the total read coverage available for the given gene. ASE data is overdispersed compared to a binomial distribution (*Castel et al., 2015*). To properly account for this overdispersion in our simulations, we used the available ASE data to estimate the overdispersion parameter $\rho$ in a beta-binomial distribution using the R function 'optim'. We provided the function with the total read count and the observed allele count for each gene and used 0.5 as the 'true' probability of success. We estimated $\rho$ separately for the Torabi and the Albert data and found that the latter was somewhat more overdispersed ($\rho = 0.0054$) than the former ($\rho = 0.0041$).

We confirmed that these overdispersion estimates fit the data reasonably well by generating simulated datasets using the observed read counts from the Albert and Torabi datasets for each gene, along with the estimated $\rho$'s. For each gene, we used the mean of the Albert and Torabi ASE fold changes to compute the target probability of success. These simulated data are meant to represent a random instance of allelic counts across genes, using distributions that closely mimic the real data. We generated 50 such datasets using the Albert and Torabi parameters, respectively. We computed all pairwise correlations between these simulated Albert and Torabi datasets. The median correlation coefficient across genes was $r = 0.42$ in the simulated data, compared to $r = 0.39$ for the observed Albert and Torabi data. Thus, while the simulations underestimate the overdispersion in the real data by a small amount, they provide a reasonable approximation.

We used these estimates of $\rho$ to simulate 1000 instances of allele counts for several combinations of total read count and fold change (*Figure 2—figure supplement 2*). We computed power as the fraction of simulations in which a binomial test yielded $p<0.05$ or $p<1.5e-5$, the Bonferroni-corrected threshold for 3340 tests. We also computed the fraction of simulations in which the direction of fold change agreed with the true change.

These simulations show that power increases with increasing read counts and effect sizes. Crucially, even with very high sequencing depth of 5000 allele-specific reads per gene, the power to detect ASE of a magnitude typical for the majority of local eQTLs detected here is less than 60%. Most genes in the ASE data have substantially lower coverage: the median coverage is ~1000 in the Torabi data and ~200 in the Albert data. Thus, we expect to miss the ASE effects of the majority of local eQTLs.

To explore this relationship more precisely, we conducted gene-matched power simulations. For each gene, we conducted 100 simulations using the estimated $\rho$ and sequencing coverage for the given ASE dataset. We used the observed eQTL fold changes to compute expected probabilities of success. We conducted these simulations separately for the Albert and Torabi data.

## Genes affected by hotspots

To estimate which genes are influenced by a given hotspot irrespective of whether these associations reached genome-wide significance for the given gene, we performed a targeted forward scan for linkage at the hotspot locations. For each chromosome and for each gene, we regressed out the effects of significant eQTLs detected on other chromosomes as well as the effects of technical covariates. Then, for each gene, we repeated the procedure described above under 'Mapping additive eQTL'. However, here X was defined to be the set of detected hotspot markers for that chromosome, as well as either the closest marker to each gene or, if a gene had a genome-wide significant local eQTL, the local eQTL peak marker for that gene. The same FDR threshold (5%) was used to identify the best model for each gene. For each gene, coefficients for the effects of all significant markers identified by this procedure were determined by multiple regression.

The regression coefficients from this model were used to count the number of genes affected by a given hotspot (i.e. all nonzero entries) and to rank genes for inclusion in gene ontology (GO) and transcription factor (TF) binding site (TFBS) enrichments, and for plotting in networks. *Source data 8* also presents the number of genes with a genome-wide significant eQTL overlapping each hotspot.

For each hotspot, TFBS and GO enrichments were calculated on the 100 genes with the strongest increases or decreases in expression from the RM allele at the hotspot, respectively. When fewer than 100 genes were influenced by the hotspot in the given direction, we used all genes influenced in this direction.

In *Source data 8*, we report GO enrichments for the genes affected by each hotspot. This table shows the top five GO categories for each hotspot that exceeded an enrichment p-value of $p \leq 0.05$ divided by the number of GO categories tested. We adjusted p-values separately for the Biological Process, Molecular Function, and Cellular Compartment GO subtrees. We did not adjust for the fact that multiple hotspots were tested, nor for the fact that two directions of effect (higher expression linked to the BY or the RM allele) were tested. *Source data 10* lists all GO enrichment results.

TFBS enrichments for genes affected by hotspots were computed using Fisher's exact test. The underlying relationships between TFs and target genes were based on regulatory relationships between genes downloaded from SGD (*Cherry et al., 2012*) on May 1$^{st}$, 2016. In *Source data 8*, we show all TFs with significant enrichments at a threshold of $p \leq 1.3e\text{-}6$, corresponding to a Bonferroni-corrected p-value controlling for 38,556 tests (189 TFs, 102 hotspots, and two directions of effect per hotspot). *Source data 8* also shows the most significant TFBS enrichment, irrespective of whether this enrichment was significant after multiple testing or not. *Source data 11* lists all TFBS enrichment results.

We displayed the genes that are affected by each hotspot in a manner that reflects significant correlations between the genes' expression levels in order to highlight groups of genes that may be functionally related. Because expression levels will become correlated when they are linked to the same hotspot, we wanted to estimate the underlying correlation matrix in a manner that is as free as possible from correlations induced by the hotspots. We fit a sparse conditional Gaussian graphical model (sCGGM [*Zhang and Kim, 2014*]) to the expression data and the genotypes at the hotspot markers. sCGGM decomposes the total correlation matrix among gene expression levels into 'direct' effects of the genetic markers on each gene expression level as well as a matrix $\theta_{yy}$ of 'indirect' effects that genes exert on each other. We fit the model on a multi-core processor using an R installation compiled using the Intel Math Kernel Library to speed up linear algebra operations, as provided by the University of Minnesota Supercomputing Institute. Following preliminary cross-validation on a subset of 2000 genes, we set the sCGGM regularization parameters $\lambda_1$ and $\lambda_2$ to 0.1.

We fit log2(TPM) values for all 5720 expressed genes. Prior to fitting, effects of experimental batch and yeast culture optical density were removed using a linear model. We also removed the effect of the marker closest to each gene. This latter correction was not performed for genes that reside in the window tested by bootstrapping around each hotspot, and for genes with a local eQTL that overlaps the given hotspot. We chose to keep the local effects for these genes because local eQTLs at these genes may underlie the hotspot, and we were interested in preserving such potential 'direct' local effects for our visualizations.

We used the entries of $\theta_{yy}$ to generate the network plots in *Figure 4—figure supplement 4* and *Supplementary file 6*. In spite of the shrinkage imposed by the sCGGM algorithm, very few entries were estimated to be zero. As a practical threshold for plotting, we excluded entries of $\theta_{yy}$ with absolute values less than 1e-5.5. This threshold was set based on visual inspection of a histogram of all entries in $\theta_{yy}$, which showed a bimodal distribution with a clear peak of values exceeding this threshold separated from a peak centered on much smaller values. Network plots were generated using the R igraph package (*Csárdi and Nepusz, 2006*). *Supplementary file 6* shows the resulting network plots for all hotspots.

## Analysis of genes located in hotspots

Several hotspots were located close to chromosome ends. Yeast chromosome ends contain complex structural variation that segregates among isolates and influences traits (*Cubillos et al., 2011*). In some cases, BY and RM differ for the presence of entire subtelomeric blocks of genes (*Bergström et al., 2014*). When a hotspot arises from these regions, the identity of the causal gene cannot be determined using our present segregant panel because each segregant either carries all

or none of the genes in these regions. Further, the marker map we used for mapping stops at the borders of these regions. Therefore, these hotspots often have very sharp bootstrap distributions on the first or last marker of the linkage map on the given chromosome. We excluded subtelomeric hotspots from the analyses of genes located in hotspots because the position of the final marker on a chromosome is unlikely to reflect the position of the causal gene, which may well be located distally to the marker. We excluded 13 hotspots whose peak marker is within 5 kb of the end of our linkage map. We focused the remaining analyses of hotspot genes on 26 non-subtelomeric hotspots with confidence regions that contain three or fewer genes, for a total of 58 genes.

To analyze features of genes located in hotspots, we performed multiple logistic regression. The dependent variable indicates whether or not a gene resides in one of these hotspots, and the set of gene features described above served as potential predictor variables. Significance tests were performed using likelihood ratio tests for dropping each term from a full additive model without interactions, as implemented in the R car package.

Gene ontology analysis on these genes was conducted using Fisher's exact test in topGO (*Alexa et al., 2006*). Genes in the yeast genome tend to be clustered in co-localized groups with similar functions (*Hurst et al., 2004*). To test if this clustering influences our GO analyses of genes located in hotspots, we performed a randomization analysis. We sampled 1000 sets of 58 neighboring genes that mirror the distribution of the number of genes across the 26 hotspots. For each set, we performed the same GO enrichment analysis as for the actual 58 hotspot genes. Within each set, we counted the number of significant (at $p < 0.05$) GO terms and used the fraction of this distribution that matched or exceeded the number observed in the real set as an empirical p-value for whether the enrichment was globally significant. This test showed significantly more enriched terms than randomly expected for the GO Biological Process category ($p = 0.001$), but only a marginal excess for Molecular Function ($p = 0.060$) and no significant excess for Cellular Compartment ($p = 0.695$).

This analysis is conservative because in the real data we considered GO terms at much more stringent p-value cutoffs than $\leq 0.05$. We further explored the FDR for GO term enrichment at more stringent p-values by dividing the mean number of terms significant at a given threshold across the permutations by the number of significant terms observed in real data. We found that for Biological Process, FDR was $< 0.05$ for GO enrichments with $p < 0.005$, which includes all terms described in the paper.

Finally, we computed an empirical p-value for each GO term by asking how often its observed p-value is matched or exceeded in the permutations. This analysis controls for different sizes and compositions of the different GO terms. All terms reported to be significant in the text had $p < 0.001$ in these analyses; enrichments as strong or exceeding the observed ones were never seen in 1000 random gene sets. We conclude that our GO analysis of genes in hotspots is unlikely to reflect random sampling of genomic regions.

Plots of hotspot location and gene content were generated using the R package Gviz (*Hahne and Ivanek, 2016*). *Supplementary file 5* shows plots of gene content for all hotspots.

## Comparisons to pQTLs

We used pQTLs for 160 proteins identified in the BY/RM cross (*Albert et al., 2014b*). The pQTL coordinates were mapped from the sacCer2 to the sacCer3 genome using the UCSC liftover tool (https://genome.ucsc.edu/cgi-bin/hgLiftOver).

The pQTLs were mapped using bulk-segregant analysis (BSA) in large pools of segregants where each gene was tagged with green fluorescent protein (*Albert et al., 2014b*). BSA does not produce effect sizes in units of gene expression levels or variance. We instead used the allele frequency difference between high and low GFP pools at the pQTL peak position as a measure of pQTL effect size. For eQTLs, we used the coefficient of the correlation between scaled expression levels and marker genotype. We chose this effect size measure because it, like the BSA allele frequency estimates, is bounded by −1 and 1, resulting in more easily interpretable scatterplots. Using other measures of eQTL effect such as multiple regression coefficients did not change our conclusions about pQTL overlap.

For the comparison of strong eQTLs to significant pQTLs, we defined 'strong' eQTLs as those eQTLs that explained $\geq 3.5\%$ of phenotypic variance. Our eQTL data had $\geq 99\%$ power to detect such eQTLs. Under the assumption that the bulk-segregant based pQTL data had similar statistical power as the current eQTL data, these eQTLs should be easily detectable as pQTL if their effects on

proteins are similarly strong. For the comparison of strong pQTLs to significant eQTLs, we defined 'strong' pQTLs as pQTLs with LOD $\geq 15$. This threshold was chosen to pick a set of strong pQTLs that had similar size (218 pQTLs) to the set of strong eQTLs (238 eQTLs).

For eQTLs that did not overlap a significant pQTL or vice versa, we used the effect size point estimate at the respective peak position in the non-significant dataset for plotting in *Figure 6B and D* and for results presented in *Supplementary files 8* and *10*. Data underlying these analyses is available as *Source data 13*.

To compute $\pi_1$ at eQTL positions in X-pQTL data, we summed counts for 21 SNPs centered on the given eQTL marker for the BY and RM allele in the high and low GFP populations (*Albert et al., 2014b*), performed a G-test on these summed counts, and analyzed the resulting p-values using the qvalue package in R (*Storey and Tibshirani, 2003*). We chose to sum counts from multiple SNPs because in X-QTL data with relatively low sequencing coverage, the counts at any one SNP are subject to counting noise. For reference, $\pi_1$ computed based only at the nearest SNP to each eQTL was 0.14 for the comparison using all eQTLs, compared to 0.66 using the 21 SNP window. We computed $\pi_1$ on 5 sets of randomly selected genome positions. These random $\pi_1$ values were all $\leq 0.52$. These high random estimates reflect the high density of X-pQTLs in the genome as well as a potential upward bias due to our choice of SNP window, but are all clearly lower than the observed value.

To compute $\pi_1$ at pQTL positions in eQTL data, we performed t-tests comparing the normalized expression levels for segregants with the BY allele at pQTL markers to those with the RM allele. The resulting p-values were analyzed using the qvalue package (*Storey and Tibshirani, 2003*).

## Genetic interactions

For each transcript with at least one significant additive eQTL, we fit a model that included the batch and growth covariates, the significant additive eQTLs, and a random effect for polygenic background. The residuals from this model were used for the detection of eQTL-eQTL interactions.

The set of markers used for the detection of eQTL-eQTL interactions was reduced to 3106 using the findCorrelation function in the R package caret (https://cran.r-project.org/web/packages/caret/index.html), using a cutoff of 0.99. All unique combinations of markers were tested for each transcript, with the exclusion of the 20 closest markers on the same chromosome. eQTL-eQTL peaks, including the rare case of closely linked eQTL-eQTL interactions occurring on the same chromosome pair, were identified using custom code provided in our code repository. The same procedure was repeated for five random permutations of segregant identities. A false discovery rate was calculated as the ratio of expected to observed peaks at different LOD thresholds. A false discovery rate for the marginal scan was calculated as the ratio of expected to observed peaks at different LOD thresholds for the subset of marker pairs where one of the pairs had a significant additive effect. False discovery rate was controlled at 10% for both scans.

Additionally, we tested a model of eQTL-eQTL interactions between significant additive eQTLs only. For each gene, we regressed out the effects of significant additive eQTLs, the effect of polygenic background, and the effects of technical covariates. Then, for each gene we tested only the eQTL-eQTL interaction effect between significant additive eQTL markers. This procedure involved the peak markers detected in the section on eQTL mapping without any marker downsampling. An F-statistic was calculated for each test. The same procedure was repeated 10 times with permutations of segregant identities. From the permutations, the expected number of significant eQTL-eQTL linkages was calculated at various thresholds. False discovery rate was controlled at 10% for this procedure. *Source data 14* lists all identified genetic interactions.

Circle plots were generated using the ggBio package in R (*Yin et al., 2012*). Pair markers were deemed to overlap an additive eQTL hotspot if they were within 10 kb of the given hotspot interval.

## Supplementary discussion

### Supplementary discussion 1 – Genes with many eQTLs

Some genes were affected by a large number of eQTLs. The five genes with the maximum identified number of 21 eQTLs included the gene *OCH1* as well as two pairs of genes that are physically located right next to each other, respectively. In each pair, the upstream gene (*NIT1* and *AQY2*, respectively) is annotated as a protein coding gene, while the downstream genes (YIL165C and YLL053C, respectively) is annotated as a 'putative' protein. In several yeast strains other than the

reference strain (a version of which is the BY strain we use here), these gene pairs each form a single open reading frame (ORF). In BY, this ORF is interrupted by a premature stop codon, resulting in truncated ORF annotations. The fact that BY tolerates the presence of a premature stop variant in these genes suggests that these genes are under low evolutionary constraint, which may help explain why these genes are also tolerant of regulatory influences from a large number of eQTLs.

## Supplementary discussion 2 – Comparison of heritability to various gene features

We asked whether the heritability for each gene was correlated with various gene characteristics. Heritability was positively correlated with expression level (*Figure 1—figure supplement 2*; *Supplementary file 1*). This result may in part be caused by higher power in more highly expressed genes. Therefore, we controlled for expression level in a multivariate analysis that tested the correlation of various gene features with heritability while controlling for all other features, including gene expression level.

Genes with higher heritability were less likely to be essential and had fewer protein-protein interaction and synthetic genetic interaction partners (*Supplementary file 1*). This suggests that highly connected 'hub' genes with many interaction partners may be less tolerant of regulatory variation. The amino acid sequence of genes with higher heritability evolved more slowly, perhaps reflecting higher sequence conservation.

Genes with higher heritability were enriched (T-test of heritability of genes in GO group vs. genes not in GO group as implemented in topGO (*Alexa et al., 2006*) for biological processes involved in mitochondrial function and cellular respiration (e.g. GO:0055114 'oxidation-reduction process': p<1e-30; GO:0045333 'cellular respiration': p=5e-8). In yeast, respiration is an optional and highly regulated means of energy production, and it is conceivable that the respiratory machinery provides a disproportionally large target for regulatory genetic variation.

Genes with lower heritability were enriched for processes involved in general biogenesis (e.g. GO:0042254 'ribosome biogenesis' p<1e-30, GO:0010467 'gene expression' p=2e-25, and GO:0000278 'mitotic cell cycle' p<1e-30), suggesting that these cellular processes may be less tolerant of regulatory variation.

The 28 genes with the highest heritability of at least 0.9 were strongly enriched for genes involved in yeast mating. For example, they included five out of seven genes annotated as 'regulation of mating-type specific transcription, DNA-templated' (GO:0007532, p=4e-11). The yeast mating type pathway involves the **a** and alpha mating types that are determined by alternative alleles at the mating type locus. Each mating type expresses a set of highly abundant genes that are almost completely shut off in the other mating type. Our mapping population includes both mating types at equal frequency. Genes involved in mating are thus expected to fall into two genetically determined groups in which their expression is either very high or nearly absent, resulting in high heritability across the segregant population.

## Supplementary discussion 3 – Relationship of eQTL number and heritability

The number of eQTLs that influenced a given gene was correlated with heritability (r = 0.56, p<2.2e-16, *Figure 1—figure supplement 2B*), as expected if each additional eQTL adds to the genetic variance. However, among genes with the highest heritability (423 genes with $h^2 \geq 0.6$), heritability was negatively correlated with the number of eQTLs (r = −0.47, p<2.2e-16; *Figure 1—figure supplement 2B*). For these genes, the strongest eQTL accounts for a progressively larger fraction of heritability (r = 0.56, p<2.2e-16, *Figure 1—figure supplement 2C & D*), while for genes with lower heritability this relationship was slightly negative (r = −0.04, p=0.001). Thus, while the lower heritability typical of most genes tended to arise from multiple eQTLs that each had small to intermediate effect, high heritability tended to arise from single, strong eQTLs. The single eQTLs that by themselves generated heritability of $\geq 0.9$ were all local eQTLs for genes in regions of the genome with high, structurally complex variation (e.g. the *ENA* locus (*Treusch et al., 2015*) or subtelomeres [*Bergström et al., 2014*]), as well as the gene *HO*, which is deleted in our RM but not in the BY strain.

## Supplementary discussion 4 – Allele-specific expression analyses

To estimate the fraction of local eQTLs that act in *cis* vs. in *trans*, we compared the local eQTLs to RNA-Seq data from a diploid BY/RM hybrid. In the hybrid, *trans*-acting genetic variation influences both alleles at a gene similarly. By contrast, *cis* effects can be detected as an allelic imbalance in expression (also called 'allele specific expression', ASE) between the BY and the RM allele. We analyzed two independent BY/RM hybrid datasets and quantified ASE for 3340 genes that have at least one variant in their coding sequence. Of these, 1974 (59%) had a genome-wide significant local eQTL.

Genes with significant ASE (Bonferroni-corrected p<0.05) had more local eQTLs than expected by chance (at least one significant ASE dataset: 451/598 genes, p<2.2e-16, odds ratio (OR)=2.5; both ASE datasets significant: 100/121; p=3e-8, OR=3.4). Most genes with ASE but without a local eQTL had ASE of very small magnitude. For these genes, the local eQTLs that would be expected to correspond to the ASE may have been missed in spite of high power in our current dataset. The remaining genes with ASE but without a local eQTL were located very close to strong eQTLs that may have been misclassified as distant (*Supplementary file 2* and *Figure 2—figure supplement 1A and B*). Overall, significant ASE typically resulted in a detectable local eQTL.

We next examined what fraction of the local eQTLs arises in *cis* from ASE vs. from local *trans* acting variation. A simulation analysis of the diploid hybrid data showed that even if there is perfect agreement between ASE and local eQTLs, the statistical power to detect ASE of the magnitude typical for most local eQTLs is limited (Materials and Methods, *Figure 2—figure supplement 2*). Indeed, 77% (1523/1974) of the local eQTLs did not have significant ASE. Those local eQTLs that had significant ASE had larger effects than those that did not (*Figure 2—figure supplement 1C*). Of the 35 local eQTLs for which power to detect ASE was at least 80%, 30 had ASE in at least one ASE dataset (26 in both datasets; *Figure 2—figure supplement 1D*). Thus, most cases of missing ASE were probably due to low statistical power, rather than caused by local *trans* effects. We present genes with the strongest discrepancies between ASE and local eQTLs in *Supplementary files 2, 3 and 4*.

Regulatory sequence variants in upstream regulatory regions such as the promoter are expected to act in *cis* and should result in both ASE and local eQTLs. Genes vary in the number of variants in their upstream regions. While the number of upstream variants was only weakly correlated with ASE (Spearman's rho = 0.04, p=0.01), there was a stronger correlation with the fold changes of local eQTLs (rho=0.23, p<2.2e-16). This analysis is imperfect because not every upstream variant has effects on expression, and because multiple variants with opposite effects could cancel each other's effect on expression. Therefore, *cis* effects are not expected to be a simple function of upstream variant number. Nevertheless, better correlation of variant number with local eQTLs than with ASE is consistent with *cis* regulatory effects that were better detected in the current well-powered eQTL data than in the available ASE data.

The absolute values of the fold changes agreed remarkably well between ASE and local eQTLs. The standardized major axis (SMA) slope for all local eQTL effects compared to ASE was 0.94 ($r^2$=0.45, p<2.2e-16). Thus, the effects of *cis*-acting variation on allelic expression were typically carried forward to local eQTLs of nearly the same magnitude. The SMA slope was just less than one (confidence interval 0.91–0.97), which may indicate a small tendency for local *trans*-acting variation to buffer some *cis*-acting variants (*Bader et al., 2015*).

## Supplementary discussion 5 – Comparison of eQTLs and protein QTLs (pQTLs)

The main text presents a comparison focused on the strongest distant eQTLs and pQTLs. The results for all genome-wide significant distant QTLs, as well as for local QTLs are presented here.

The 154 genes that were present in both our current dataset and our earlier X-pQTL data (*Albert et al., 2014b*) had 1,059 distant eQTLs and 1,024 distant pQTLs. Of these eQTLs, 30% (321) overlapped a pQTL, while of the pQTLs, 31% (314) overlapped an eQTL. The number of overlapping QTLs differed between these two comparisons because a QTL in one dataset can overlap two neighboring QTLs in the other dataset due to wide confidence intervals of weaker QTLs. Of the overlapping QTLs, 77% (254 / 331) had the same direction of effect (*Figure 6C*). π1 statistics suggested that 66% of all distant eQTLs have a matching pQTL, while 47% of all distant pQTLs match an eQTL. These estimates for overlap among all distant QTLs are lower than those for strong QTLs reported

in the main text. This is expected, given the small effect sizes of most distant QTLs will frequently be missed even when comparing two datasets with high statistical power.

Our X-pQTL dataset comprised experiments for testing the effects of local pQTLs for 41 genes (*Albert et al., 2014b*), which revealed local pQTLs for 21 genes (at a LOD of >3). From the current mRNA data, we observed local eQTLs for 27 of these 41 genes. We found both a local pQTL and a local eQTL for 16 genes, which was not more than expected by chance (one-sided Fisher's exact test, p=0.14, odds ratio=2.6). However, there was a positive correlation between effect sizes for local eQTLs and pQTLs for all 41 genes (Spearman's rho=0.6, p=0.0001). Tests for local pQTLs with high statistical power at more genes will be necessary to draw firm conclusions about the agreement of local eQTLs and pQTLs.

# Acknowledgements

We thank Noorossadat Torabi for unpublished ASE data, Meru Sadhu and Olga Schubert for comments on the manuscript, and the Minnesota Supercomputing Institute (MSI) at the University of Minnesota for computational resources.

# Additional information

### Competing interests

Leonid Kruglyak: Reviewing editor, *eLife*. The other authors declare that no competing interests exist.

### Funding

| Funder | Grant reference number | Author |
| --- | --- | --- |
| Howard Hughes Medical Institute | | Leonid Kruglyak |
| National Institute of General Medical Sciences | R01GM102308 | Leonid Kruglyak |
| National Institute of General Medical Sciences | 1R35GM124676-01 | Frank Wolfgang Albert |

The funders had no role in study design, data collection and interpretation, or the decision to submit the work for publication.

### Author contributions

Frank Wolfgang Albert, Joshua S Bloom, Conceptualization, Resources, Data curation, Software, Formal analysis, Validation, Investigation, Visualization, Methodology, Writing—original draft, Project administration, Writing—review and editing; Jake Siegel, Investigation, Methodology; Laura Day, Investigation, Collected a substantial portion of the data; Leonid Kruglyak, Conceptualization, Resources, Supervision, Funding acquisition, Writing—original draft, Project administration, Writing—review and editing

### Author ORCIDs

Frank Wolfgang Albert  http://orcid.org/0000-0002-1380-8063
Joshua S Bloom  http://orcid.org/0000-0002-7241-1648
Leonid Kruglyak  https://orcid.org/0000-0002-8065-3057

### Decision letter and Author response

Decision letter https://doi.org/10.7554/eLife.35471.061
Author response https://doi.org/10.7554/eLife.35471.062

# Additional files

## Supplementary files

• Source data 1. Expression values. This xlsx file contains expression levels in units of log2(TPM) for all genes and segregants.
DOI: https://doi.org/10.7554/eLife.35471.021

• Source data 2. Covariates. This xlsx file contains information on experimental batch and growth covariates for all segregants.
DOI: https://doi.org/10.7554/eLife.35471.022

• Source data 3. Genotypes. This xlsx file contains genotypes at 42,052 markers for all segregants. BY (i.e. reference) alleles are denoted by '−1'. RM alleles are denoted by '1'.
DOI: https://doi.org/10.7554/eLife.35471.023

• Source data 4. Detected eQTLs. This xlsx file contains all detected eQTLs.
DOI: https://doi.org/10.7554/eLife.35471.024

• Source data 5. Heritability estimates. This xlsx file contains additive and pairwise interactive genetic variance estimates for all genes.
DOI: https://doi.org/10.7554/eLife.35471.025

• Source data 6. Gene characteristics and annotations. This xlsx file contains gene features used for comparisons with heritability.
DOI: https://doi.org/10.7554/eLife.35471.026

• Source data 7. Data for allele-specific expression comparisons. This xlsx file contains processed data used to compare ASE and local eQTLs.
DOI: https://doi.org/10.7554/eLife.35471.027

• Source data 8. Overview of eQTL hotspots. This xlsx file presents information on the detected hotspots such as position, number of genes affected, genes located in the region, GO and transcription factor binding site enrichments of the affected genes. The file has a legend in a separate work sheet.
DOI: https://doi.org/10.7554/eLife.35471.028

• Source data 9. Hotspot effect matrix. This xlsx file contains estimated effects of all hotspots on all genes.
DOI: https://doi.org/10.7554/eLife.35471.029

• Source data 10. GO enrichment results for genes affected by each hotspot. This txt file contains GO enrichment results for the 102 hotspots in one file. Columns indicate hotspot identity, GO subcategory ('biological process', 'molecular function', or 'cellular compartment'), and the group of genes tested for enrichment ('RM'=up to 100 genes with the strongest hotspot effects with higher expression linked to the RM allele, 'BY'=as for 'RM', but higher expression linked to the BY allele).
DOI: https://doi.org/10.7554/eLife.35471.030

• Source data 11. Enrichments for transcription factor binding sites at genes affected by each hotspot. This txt file contains enrichment results for the 102 hotspots in one file. Columns indicate hotspot identity, and the group of genes tested for enrichment ('RM'=up to 100 genes with the strongest hotspot effects with higher expression linked to the RM allele, 'BY'=as for 'RM', but higher expression linked to the BY allele).
DOI: https://doi.org/10.7554/eLife.35471.031

• Source data 12. GO enrichment results for genes located in hotspots. This xlsx file contains the results of a GO enrichment analysis of genes physically located in hotspot regions.
DOI: https://doi.org/10.7554/eLife.35471.032

• Source data 13. Comparison with pQTLs. This xlsx file contains processed data used for the comparisons of eQTLs and pQTLs.
DOI: https://doi.org/10.7554/eLife.35471.033

• Source data 14. Detected interactions between eQTL pairs. This xlsx file contains all detected pairs of eQTLs with non-additive genetic interactions. There are separate sheets for the genome-wide scan, the scan between additive eQTLs and the genome, and the scan between additive eQTLs. There is also a legend in a separate sheet.
DOI: https://doi.org/10.7554/eLife.35471.034

• Supplementary file 1. Multiple regression of heritability on various gene features. Sums of squares, degrees of freedom and F-values were computed using Type II analysis of variance as implemented in the R car package. (1) log2(TPM).
DOI: https://doi.org/10.7554/eLife.35471.035

• Supplementary file 2. Genes with strong (more than 2-fold) and significant ASE in both datasets but no local eQTL. (1) Shown is the less significant p-value from the two ASE datasets. (2) The LOD score at the gene position itself irrespective of whether this eQTL is significant. (3) Positive values indicate higher expression in RM compared to BY. (4) These genes have strong eQTLs close to the gene, but with a confidence interval that just excludes the gene. The may be influenced by *cis* acting local eQTLs where the causal variant is located further away from the gene than captured by our definition of upstream regulatory regions as 1000 base pairs upstream of the start codon.
DOI: https://doi.org/10.7554/eLife.35471.036

• Supplementary file 3. Genes with a local eQTL but no ASE in spite of $\geq$80% power to detect ASE. (1) Positive values indicate higher expression in RM compared to BY. (2) Shown is the more significant p-value from the two ASE datasets. (3) The nominally significant p-values in this column do not pass Bonferroni cutoff for significance. Therefore, ASE at these genes was not identified as significant.
DOI: https://doi.org/10.7554/eLife.35471.037

• Supplementary file 4. Genes with a local eQTL and significant ASE, and discordant direction of effect. (1) Positive values indicate higher expression in RM compared to BY. (2) Shown is the less significant p-value from the two ASE datasets. (3) The table shows only genes where both ASE datasets agreed in the direction of effect. Shown is the average effect.
DOI: https://doi.org/10.7554/eLife.35471.038

• Supplementary file 5. Hotspot region plots. This pdf file contains one page per hotspot region, showing the position, bootstrap distribution, variant and gene content for each hotspot. See legend of *Figure 4* and *Figure 4—figure supplement 4* for further details.
DOI: https://doi.org/10.7554/eLife.35471.039

• Supplementary file 6. Plots of genes affected by hotspots. This pdf file contains one page per hotspot region. Each page shows three gene co-expression networks. Each network displays connections among genes with the strongest linkages to the hotspot. The left plots show genes with higher expression in the presence of the RM allele. The middle plot shows genes with higher expression in the presence of the BY allele. The left and middle panels each show up to 50 genes with the strongest linkages to the hotspot in the given direction. If fewer than 50 genes were affected by the hotspot in the given direction, all genes affected in this direction are plotted, and their number indicated in the panel title. The right panel shows the joint network formed by the genes in the left and middle panel, with colors indicating up- or down-regulation by the hotspot. Genes with local eQTL located in the 95% confidence interval for hotspot location are indicated by red circles. Genes local eQTL located anywhere in the region tested by bootstraps are indicated by orange circles. See legend of *Figure 4—figure supplement 4* for further details.
DOI: https://doi.org/10.7554/eLife.35471.040

• Supplementary file 7. Multiple logistic regression of genes located in hotspots on various gene features. Likelihood ratios, degrees of freedom and p-values were computed using Type II analysis of variance as implemented in the R car package. (1) log2(TPM).
DOI: https://doi.org/10.7554/eLife.35471.041

• Supplementary file 8. Strong eQTLs without pQTL. (1) Positive values indicate higher expression in RM compared to BY.
DOI: https://doi.org/10.7554/eLife.35471.042

• Supplementary file 9. Strong mRNA and protein QTLs with opposite effect. (1) Positive values indicate higher expression in RM compared to BY.
DOI: https://doi.org/10.7554/eLife.35471.043

• Supplementary file 10. Strong pQTLs without eQTL. (1) Positive values indicate higher expression in RM compared to BY.
DOI: https://doi.org/10.7554/eLife.35471.044

• Transparent reporting form

DOI: https://doi.org/10.7554/eLife.35471.045

## Data availability

Sequencing reads have been deposited at SRA under accession codes SRP148919 and SRP149494. Processed datasets are included with this manuscript, and are also available in a FigShare repository at https://figshare.com/s/83bddc1ddf3f97108ad4. All data are available freely and without restriction.

The following datasets were generated:

| Author(s) | Year | Dataset title | Dataset URL | Database, license, and accessibility information |
|---|---|---|---|---|
| Albert FW | 2018 | Sequencing reads from Genetics of trans-regulatory variation in gene expression | https://www.ncbi.nlm.nih.gov/sra/SRP148919 | Publicly available at the NCBI Sequence Read Archive (accession no: SRP148919) |
| Torabi N, Kruglyak L | 2012 | Sequencing reads from a yeast diploid BY/RM strain | https://www.ncbi.nlm.nih.gov/sra/SRP149494 | Publicly available at the NCBI Sequence Read Archive (accession no: SRP149494) |
| Frank Wolfgang Albert, Joshua S Bloom, Jake Siegel, Laura Day, Leonid Kruglyak | 2018 | Data for Albert, Bloom, et al, "Genetics of trans-regulatory variation in gene expression" | https://figshare.com/s/83bddc1ddf3f97108ad4 | Available at figshare under a CC0 Public Domain licence (https://figshare.com/). |

The following previously published datasets were used:

| Author(s) | Year | Dataset title | Dataset URL | Database, license, and accessibility information |
|---|---|---|---|---|
| Albert FW, Treusch S, Shockley AH, Bloom JS, Kruglyak L | 2014 | Genetics of single-cell protein abundance variation in large yeast populations | https://www.nature.com/articles/nature12904 | Supplementary Data 2 and 3 |
| Albert FW, Muzzey D, Weissman JS, Kruglyak L | 2014 | Genetic Influences on Translation in Yeast | http://journals.plos.org/plosgenetics/article?id=10.1371/journal.pgen.1004692 | Data S2 |
| Costanzo | 2016 | A global genetic interaction network maps a wiring diagram of cellular function | http://thecellmap.org/costanzo2016/ | Raw genetic interaction datasets: Pair-wise interaction format |
| Bloom JS, Ehrenreich IM, Loo WT, Lite TLV, Kruglyak L | 2013 | Finding the sources of missing heritability in a yeast cross | http://genomics-pubs.princeton.edu/Yeast-Cross_BYxRM/data.shtml | Genomic DNA sequencing reads |

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
