## [Decision Letter]

Thank you for submitting your article "Genetics of *trans*-regulatory variation in gene expression" for consideration by *eLife*. Your article has been reviewed by three peer reviewers, and the evaluation has been overseen by a Reviewing Editor and Patricia Wittkopp as the Senior Editor. The reviewers have opted to remain anonymous.

The reviewers have discussed the reviews with one another and the Reviewing Editor has drafted this decision to help you prepare a revised submission.

Summary:

This manuscript makes use of an elegant experimental design to investigate how genetic variation influences gene expression in *cis* and *trans*. Using over a thousand genotyped segregants from a cross between two yeast strains, the authors investigate how inherited genetic variation accounts for differences in gene expression levels as measured by RNA-seq. The eQTL catalogue is fairly exhaustive and explains over 70% of the heritability in gene expression levels. The authors report that while 50% of genes harbor local, relatively large-effect *cis*-eQTLs, the aggregate effect of *trans*-acting eQTLs typically outweigh that of local variation. Those *trans*-eQTLs cluster in a small number of hotspots that are responsible for the majority of gene expression variation, likely due to reverberating indirect effects through the cell machinery. The study involves a large amount of experimental work and seems to have been designed with great care.

Concerns were raised about the novelty of many of the findings in this study relative to prior work, but the reviewers and editor were ultimately swayed to support publication based on the advance in rigor and resolution provided by this work combined with some new findings for the field. There were also other issues raised, however, that we think can be successfully addressed in a revision within the next 2 months.

Essential revisions:

1) Figures are quite poorly designed. Figure 5 is a particularly egregious example: panel B would be more readable as a violin plot, and there is definitely a better way to present panel A than grey lines of variable width with no legend. Color heatmaps such as used for chromosome contacts data come to mind. Similar concerns stand for Figure 1B and D, Figure 2A and Figure 4.

2) Most of the biological interpretation on *trans*-eQTLs hotspots relies on Gene Ontology analyses, and is rather weak and speculative. Considering that these hotspots contain variants with the ability to reverberate through the gene expression network and influence many genes, a more thorough analysis would have been appreciated: are some transcription factor binding sites overrepresented in those hotspots overall? How do these hotspots relate to the 3D structure of the genome? Do they exhibit special features relative to gene organization? During our discussion, we were unable to identify a particular set of analyses that would best address this point, however, so are passing along this comment for your consideration only.

3) The analysis on eQTLs with epistatic effects fell very short considering that, as the authors note, this study design is one of the few that can identify such interactions. Please expand the discussion and, if necessary, analyses in this section.

4) Relating these findings in yeast to human cells may be more complicated than suggested in the text. Please either better justify or tone down this extrapolation. Some differences articulated by one of the reviewers:

Much throughout the text, the authors draw links between the genetic architectures of gene expression levels in yeasts and in humans. While there are indeed similarities in the numbers, e.g. the median heritability of gene expression levels, or *trans*-variants accounting for 2-3X more expression variation than *cis*-variants, I do not think that there is enough evidence that the conclusions and observations made in this work applies to human cells (which I felt was implicit in the text). In fact, I do not think that the observations here apply to humans:

For example, there is limited to no evidence from current human data that *trans*-eQTLs cluster in a very small number of genomic regions. Even with the few number of *trans*-eQTLs that were found in data from GTEx (100-500 per tissue, I believe?), we would expect to see significant clustering. I am not sure if anyone has looked into this or seen this? The authors write that "the limited number of human *trans*-eQTL discovered to date also tend to influence the expression of multiple genes […] suggesting that a similar hotspot-dominated architecture could underlie human expression variation" but I find that quite unconvincing.

From Figure 1C, it seems that distant (*trans*) eQTLs explain typically 0.5-1X as much gene expression level variation compared to *cis*-eQTLs. I believe that in humans, the difference is much larger, i.e. *trans* eQTLs explain only 0.01 times to 0.05 times as much gene expression level variation as *cis*-eQTLs. Because the *trans* vs *cis* contributions are similar between humans and yeast, this could imply that *trans*-eQTLs for human genes are distributed far more uniformly than for yeast genes.

Furthermore another difference is that we would expect a strong enrichment for coding variants for the human trans-eQTLs that were found in previous studies, which does not seem to be case. Might there be strong selection against trans-variants with ubiquitous effects in humans?

For these reasons, I would suggest either the author to provide evidence that their conclusions apply to human, or explicitly caution readers against this interpretation.

---

## [Author Response]

Essential revisions:1) Figures are quite poorly designed. Figure 5 is a particularly egregious example: panel B would be more readable as a violin plot, and there is definitely a better way to present panel A than grey lines of variable width with no legend. Color heatmaps such as used for chromosome contacts data come to mind. Similar concerns stand for Figure 1B and D, Figure 2A and Figure 4.

We have made substantial changes to the figures as follows:

Figure 5 (the reviewers’ main concern) has been completely revised and replaced with four figures (9–12), as part of our expanded treatment of epistatic interactions. Specifically, these figures show:

Figure 9: Locations and examples of markers that form epistatic interactions in *trans.*

Figure 10: Locations and examples of markers that form *cis* by *trans* epistatic interactions.

In Figures 9 and 10, we have chosen to represent interactions as circle plots. We have explored visualizations using heatmaps as suggested by the reviewers but decided against this representation because the interactions are relatively sparse, so that heatmaps are dominated by uninformative white space.

Figure 11: A revised version of the former Figure 5B, showing the distribution of effects from additive eQTLs vs. pairwise interactions. This figure now incorporates a violin plot, following the reviewers’ suggestions.

Figure 12: Examples of interactions without detectable additive effects.

Figure 1B has been altered by changing the alpha value of the points to make the shape of the center of the cloud more visible. We have also added a new panel 1D showing the distribution of heritability explained across all genes. We believe that these revised panels, 1B and 1D, completely represent this important information.

The former Figure 1D has been replaced by a new Figure 2. Panel 2B is the former inset of Figure 1D, while the new panel 2A is much more detailed visualization of the contribution of local and distant eQTLs.

We have chosen to leave Figure 2A (now 3A) unchanged. We had carefully thought about representing the entire eQTL set in a single panel and tried multiple different visualizations for our initial submission. We believe that this figure is a vivid and informative way to display the overall result of the paper, in terms of 1) the dense diagonal of many, strong local eQTLs, 2) the highly non-uniform distribution of the *trans* eQTLs, 3) the *trans*-depleted regions we discuss later in the text, and 4) differences in effect sizes among eQTLs.

We have made subtle visual changes to Figure 4B (now 6B), and, more importantly, added an inset to Figure 6B that shows the difference between the expectation and observation for each eQTL bin.

We have chosen to leave Figure 4A (now 6A) as is. We believe that the current display is an accurate and informative representation of both the random distribution (in particular, it highlights the high number of local eQTLs per hotspot expected by chance) as well as the slight but significant enrichment in real data.

We have moved the part of former Figure S6 on the *ERC1* hotspot into the main text (now Figure 5). We also moved the two figures about eQTL/pQTL agreement into the main text (now Figures 7 and 8).

We added two figure supplements for Figure 4 (Figure 4—figure supplement 1 and Figure 4—figure supplement 2) on our new result on hotspots at transcription factors with damaging mutations.

Together, we believe these changes greatly improve the visual accessibility of our results. We thank the reviewers for their guidance to reconsider how we had displayed our results.

2) Most of the biological interpretation on trans-eQTLs hotspots relies on Gene Ontology analyses, and is rather weak and speculative. Considering that these hotspots contain variants with the ability to reverberate through the gene expression network and influence many genes, a more thorough analysis would have been appreciated: are some transcription factor binding sites overrepresented in those hotspots overall? How do these hotspots relate to the 3D structure of the genome? Do they exhibit special features relative to gene organization? During our discussion, we were unable to identify a particular set of analyses that would best address this point, however, so are passing along this comment for your consideration only.

We have carefully considered these suggestions and performed additional analyses as detailed below. In addition, we suspect that our previous manuscript may have been too laconic in pointing out the many analyses we had in fact performed on each hotspot, and which were presented in a series of supplementary data and image files. We have now added expanded text to the results drawing attention to these files, and have moved parts of former Figure S6 (now Figure 5) to the main text.

We have also performed additional analyses of transcription factor genes with damaging variants. Strikingly, we found that out of 8 such genes in our cross, 6 did reside in a hotspot, often very close to the hotspot peak. One of the remaining two cases may be explained by incorrect gene annotation in the reference genome (the variant does not in fact reside in a coding region), while the other involves a frameshift very late in the coding region, perhaps limiting its effect on gene function. We have added these analyses to the Results and Discussion and have added two figure supplements (Figure 4—figure supplements 2 and 3) displaying these results. We are grateful to the reviewers and editor for their recommendation to revisit the transcription factors enrichment.

Regarding the reviewers’ specific suggestions:

We tested if the genes in the narrow hotspot regions we use in the main text were enriched for having transcription factor binding sites (TFBS). We did not find any convincing enrichment beyond what is expected by chance. We note that no such enrichment is expected *a priori* because it is not clear why multiple, independent hotspots across the genome should happen to be targets of a given transcription factor (this is the reason we had not performed such an analysis before). We have decided not to add these analyses to the manuscript. We do stress that the target genes of each hotspot had of course been tested for TFBS enrichment, and these analyses were summarized in Supplementary file 10 and fully detailed in the Materials and methods. Our expanded exposition of these tables may help address this comment.

We also tested whether there is enrichment for chromatin contacts between hotspot regions and their target genes, using data from Duan et al., 2010 (https://www.ncbi.nlm.nih.gov/pubmed/20436457), focusing on the HindIII-dreived contacts as Duan et al. did in their paper. We counted contacts between each hotspot and its targets, and found that these closely matched the distribution expected by chance, if target genes were assigned to hotspots at random. The most significant “enrichment” was for the *IRA2* locus, which had a randomization p-value of 0.005. We note that this is no better than expected by chance given that we tested 102 hotspots.

We point out that the assumption for distant eQTL hotspots in yeast is to act in *trans*, i.e. via a diffusible factor. This has indeed been found for every hotspot for which the causal gene is known so far (including *IRA2*). Further, gene regulation in yeast does not involve distant enhancers that act by distant chromatin contacts. Yeast interphase chromatin is organized such that inter-chromosomal contacts are largely determined by chromosomal position and length, rather than regulatory contacts between distant loci (https://www.ncbi.nlm.nih.gov/pubmed/22940469). As such, no enrichment of chromatin contacts between a hotspot and its targets would be expected *a priori*. Given these considerations, we have chosen not to include these (negative and expected) results in the paper.

3) The analysis on eQTLs with epistatic effects fell very short considering that, as the authors note, this study design is one of the few that can identify such interactions. Please expand the discussion and, if necessary, analyses in this section.

We thank the reviewers for this prompt to expand the Results and Discussion sections. We have now added more detailed analyses of the epistatic pairs identified by the fully unbiased search and have added or expanded the following key observations to the paper, along with new and revised figures:

– Many interactions involve additive *trans*-hotspots (Figure 9).

– Many interactions involve a local and a distant eQTL (Figure 10).

– Most epistatic pairs involve markers that also show additive effects.

– There are hardly any instances of purely non-additive effects (Figure 12 presents the best examples).

We have expanded our Discussion accordingly, in which we stress that while epistatic interactions clearly exist, their overall contribution to expression variance is small.

We have also expanded the information given in Dataset 14, which now presents the locations as well as effects of each detected epistatic pair.

4) Relating these findings in yeast to human cells may be more complicated than suggested in the text. Please either better justify or tone down this extrapolation. Some differences articulated by one of the reviewers:Much throughout the text, the authors draw links between the genetic architectures of gene expression levels in yeasts and in humans. While there are indeed similarities in the numbers, e.g. the median heritability of gene expression levels, or trans-variants accounting for 2-3X more expression variation than cis-variants, I do not think that there is enough evidence that the conclusions and observations made in this work applies to human cells (which I felt was implicit in the text). In fact, I do not think that they the observations here applies to humans:

We certainly agree that there are many important differences between humans and yeast. It was not our intention to imply that our results would directly translate to humans. We have carefully searched the text for statements that could be understood that way and have edited them accordingly. We have also added two paragraphs to the Discussion that make explicit where we do and do not see parallels between yeast and human eQTL results, and which lessons for human eQTL studies can and cannot be drawn from our current work.

We address this reviewer’s specific comments below.

For example, there is limited to no evidence from current human data that trans-eQTLs cluster in a very small number of genomic regions. Even with the few number of trans-eQTLs that were found in data from GTEx (100-500 per tissue, I believe?), we would expect to see significant clustering. I am not sure if anyone has looked into this or seen this? The authors write that "the limited number of human trans-eQTL discovered to date also tend to influence the expression of multiple genes [...] suggesting that a similar hotspot-dominated architecture could underlie human expression variation" but I find that quite unconvincing.

A careful literature review confirmed that human *trans* eQTLs have, in fact, been reported to cluster more than expected by chance. This result has been seen beginning with the earliest well-powered searches for human *trans* eQTLs (e.g. https://www.ncbi.nlm.nih.gov/pmc/articles/PMC3150446/) and has recently culminated in evidence for “hundreds of *trans*-eQTLs [that] each affect hundreds of transcripts” (https://www.ncbi.nlm.nih.gov/pmc/articles/PMC5384037/). We acknowledge that this point had been addressed only in passing in our manuscript. We have added a review paragraph in the Discussion providing an extended, current set of human *trans* eQTL references expanding on this point.

From Figure 1C, it seems that distant (trans) eQTLs explain typically 0.5-1X as much gene expression level variation compared to cis-eQTLs. I believe that in humans, the difference is much larger, i.e. trans eQTLs explain only 0.01 times to 0.05 times as much gene expression level variation as cis-eQTLs. Because the trans vs cis contributions are similar between humans and yeast, this could imply that trans-eQTLs for human genes are distributed far more uniformly than for yeast genes.

Unfortunately, we were unable to find a reference that would support the reviewer’s specific values for the relative effects of human *trans* vs *cis* eQTL effect sizes. Nevertheless, we acknowledge the reviewer’s point, and have added explicit language to the discussion stating that human *trans* eQTLs may be more uniformly distributed than we see in our yeast cross. Resolution of this question will require further research in humans. We do find it relevant to state (and have done so in the Discussion), that early eQTL searches in this cross have revealed much sparser *trans* architectures than we now see, suggesting that human *trans* eQTLs may yet follow a similar route of discovery. We have also added text enumerating some of the reasons (more variation, more complex gene regulation, more genes, and cell-type-specificity) for which the human *trans* eQTL landscape may turn out to be even more complex than what we observed here.

Furthermore another difference is that we would expect a strong enrichment for coding variants for the human trans-eQTLs that were found in previous studies, which does not seem to be case. Might there be strong selection against trans-variants with ubiquitous effects in humans?

We are not sure why the reviewer drew a conclusion about coding (vs. non-coding?) variants from our data, given that our spatial resolution does not allow statements about individual causal variants. We have searched the text but were unable to find a statement to this effect. We wholeheartedly agree that the question of the molecular nature of causal variants underlying *trans* eQTLs is highly interesting.

We also agree that the action of selection on causal variants is an exciting topic. However, we feel that a deep exploration of this important question is beyond the scope of our current manuscript. We look forward to future analyses by the reviewer and the community of the data we introduce here.

For these reasons, I would suggest either the author to provide evidence that their conclusions apply to human, or explicitly caution readers against this interpretation.

We thank this reviewer for their thought-provoking suggestions. We believe that our expanded discussion, extended literature review, and other edits throughout the text are sufficient to address concerns about a simplistic translation of our results to humans.